# Latent Mixture of Symmetries for Sample-Efficient Dynamic Learning

**Haoran Li**   **Chenhan Xiao**   **Muhao Guo**   **Yang Weng**
School of Electrical, Computer and Energy Engineering
Arizona State University, Tempe, AZ 85281
{lhaoran, cxiao20, mguo26, yweng2}@asu.edu

## Abstract

Learning dynamics is essential for model-based control and Reinforcement Learning in engineering systems, such as robotics and power systems. However, limited system measurements, such as those from low-resolution sensors, demand sample-efficient learning. Symmetry provides a powerful inductive bias by characterizing equivariant relations in system states to improve sample efficiency. While recent methods attempt to discover symmetries from data, they typically assume a single global symmetry group and treat symmetry discovery and dynamic learning as separate tasks, leading to limited expressiveness and error accumulation. In this paper, we propose the *Latent Mixture of Symmetries (Latent MoS)*, an expressive model that captures a mixture of symmetry-governed latent factors from complex dynamical measurements. Latent MoS focuses on dynamic learning while locally and provably preserving the underlying symmetric transformations. To further capture long-term equivariance, we introduce a hierarchical architecture that stacks MoS blocks. Numerical experiments in diverse physical systems demonstrate that Latent MoS outperforms state-of-the-art baselines in interpolation and extrapolation tasks while offering interpretable latent representations suitable for future geometric and safety-critical analyses.

## 1   Introduction

Learning dynamic models is fundamental to modern physical and control systems, especially when exact governing equations are unavailable due to incomplete records, device aging, or privacy constraints [1, 2]. Accurate dynamic modeling enables robust control and Reinforcement Learning [3], and supports model-based analysis for performance and safety guarantees [4, 5]. However, sample-efficient learning remains a challenge in practice, where data scarcity arises from low-resolution meters, communication failures, or storage limitations. For example, in smart grids, SCADA (Supervisory Control and Data Acquisition) systems typically report measurements every $2 \sim 10$ seconds, while critical events may occur within milliseconds [6, 7].

To mitigate data scarcity issues, existing methods can be grouped into four main categories. The first category interpolates the low-resolution data by assuming sparsity [8], low-rankness [9], or manifold geometry [10]. The second category imposes known physical laws as constraints on the learning process [11, 12]. The third category utilizes Bayesian frameworks that demand accurate prior distributions [13–15]. In general, all of the above methods require certain domain-specific assumptions or knowledge. The fourth category requires little prior knowledge and leverages Ordinary Differential Equation (ODE) solvers to construct a continuous ODE process, e.g., the family of Neural ODEs [16–19]. Although Neural ODE-based methods are well-suited for sparse data, they often employ generic architectures such as multilayer perceptrons [16, 20]. This lack of inductive bias hinders generalization, leading to overfitting rather than capturing true dynamics.

To address these limitations, we propose to use the underlying symmetries in dynamical systems. Symmetry, defined as a group of transformations that leave a system's behavior or properties equivari-

ant [21], is prevalent across various domains. For instance, rotational, reflectional, and translational symmetries are leveraged in robotics [22], 3D vision data [23], power systems [24], fluid dynamics [25] and networked systems [26]. Why can symmetry increase sample efficiency? Intuitively, symmetry transformations enable a system state to represent a large set of equivariant states, reducing the amount of data needed for training. A rich line of work has shown the improvements of enforcing symmetry priors into equivariant neural networks [27–30] with some applications to sample-efficient RL [31–33]. Some other works investigate symmetry-based data augmentation [34] and regularization [35–37] for dynamic modeling. They rely on clear symmetry knowledge. For systems with unknown symmetries, several recent methods propose to discover symmetries from data [38–42].

However, these methods incur large modeling errors when applied to complex dynamical systems due to the following limitations: (1) *Decoupled learning and accumulated errors*. Symmetry discovery and dynamic modeling are typically treated as separate processes, leading to error propagation. (2) *Limited expressiveness of symmetry representations*. Most approaches assume that a single (linear or nonlinear) symmetry group governs at least a segment of data evolution. In practice, however, the underlying transformations exhibit substantial variability: both in *which* symmetries are relevant (often multi-modal and context-dependent), and in *when* they occur (ranging from short to long terms across distant segments). To summarize, dynamical systems often do not adhere to strict, rigid symmetry constraints, but instead exhibit forms of *symmetry breaking*. This means that while approximate symmetry structures are present, they may be only partially preserved or vary across regimes. Our work identifies symmetry breaking in two orthogonal axes: a mixture of groups and a short- or long-time dependence (see Section 2.1). The mixture of groups appears in literature [43], and the locally adaptable symmetry breaking is discussed in [44–46]. They are independently discussed, but our method addresses both in a unified framework.

To address these challenges, we introduce the following modeling strategies. First, grounded in Lie theory [47], we propose a dynamic learning model that preserves local symmetries in the latent space, thereby eliminating the need for symmetry discovery algorithms. Second, rather than relying on a single symmetry group, we consider a mixture of transformations inspired by the expressive mixture-based model [48]. We refer to the overall framework as the *Latent Mixture of Symmetries (Latent MoS)*. Latent MoS encodes system evolution as a mixture of latent flows, each governed by a distinct Lie group. This design enables sample-efficient learning, as each flow captures a local symmetry transformation that defines a neighborhood of equivariant observations. Finally, to extend beyond local equivariance, we develop a hierarchical architecture by stacking multiple MoS blocks with different temporal resolutions. Namely, a higher level of MoS is responsible for modeling longer-term symmetries. In summary, our contributions are as follows:

- **Latent MoS Framework**: We propose the *Latent Mixture of Symmetries* (Latent MoS), a novel symmetry-preserving model that achieves high sample efficiency.
- **Theoretical Foundations**: We establish guarantees on the approximation capacity of Latent MoS and its ability to preserve symmetry structures induced by Lie group transformations.
- **Empirical Validation**: We demonstrate that Latent MoS outperforms state-of-the-art models across a variety of dynamical systems, particularly in low-resolution scenarios.

## 2 Preliminaries

Let $x \in \mathcal{X} \subseteq \mathbb{R}^n$ denote the system state. The discretized dynamics of the system are given by $x(t_{i+1}) = f(x(t_i))$, where $f : \mathcal{X} \to \mathcal{X}$ is an unknown dynamic function. The objective is to learn a model that approximates $f$ using historical observations $\{x(t_i)\}_{i \in \mathcal{N}_x}$, which span the time interval from $t_0$ to $t_N$ but can be *low-resolution* and *irregularly sampled*. For systems with inherent uncertainty, the goal becomes learning the transition probability distribution $p(x(t_{i+1}) \mid x(t_i))$. For notational simplicity, we focus on the deterministic setting and use $f$ in the following derivations. In addition, the goal can also be extended to multi-horizon prediction, e.g., long-term forecasting.

### 2.1 Equivariance for Efficient Dynamic Learning

Intrinsic data scarcity challenges the learning process. Hence, we introduce the equivariance:

**Definition 1.** *Let $G$ be a symmetry group and a nonlinear group action be $\pi'_s(g, \cdot) : G \times \mathcal{X} \to \mathcal{X}$. Then, the dynamic governing function $f$ is $G$-equivariant if $\forall g \in G, x(t_i) \in \mathcal{X}$:*

$$\pi'_s(g, x(t_{i+1})) = f(\pi'_s(g, x(t_i))). \tag{1}$$

When $\pi'_s(g, \cdot)$ is known or learned, existing methods use it to augment data [34] or impose symmetry constraints as regularization during training [37], thereby improving sample efficiency. While many recent models adopt this equivariance concept [32, 40, 37], we propose the following symmetry-breaking generalizations to suit more complex dynamical behaviors:

- **Temporal Dependence**. $\pi'_s(g, \cdot)$ may depend on time and past observations, requiring the learning model to be temporally adaptive. This extension is described in Section 3.2.

- **Mixture of Groups:** Multiple symmetry groups $\{G_1, \ldots, G_K\}$, such as the orthogonal group, the similarity group, the Euclidean group, and the affine group, may simultaneously influence the system dynamics. The mixture model is discussed in Section 3.2.

- **Short- and Long-Term Equivariance:** Symmetry relations may exist across multiple temporal scales. For example, $\boldsymbol{x}(t_i)$ and $\pi'_s(g, \boldsymbol{x}(t_i))$ may correspond to states that are either temporally close or far apart, depending on the choice of $g \in G$. However, traditional symmetry discovery methods often fail to capture long-term symmetries due to limited temporal modeling. Our approach to multi-scale symmetry is discussed in Section 3.3.

## 2.2 Nonlinear Group Action Decomposition

The nonlinear $\pi'_s(g, \cdot)$ is hard to analyze. [40] provides an effective decomposition for a compact Lie group $G$ and a continuous group action $\pi'_s(g, \cdot)$ $(g \in G)$:

$$\pi'_s(g, \cdot) = f_{\text{dec}} \circ \pi_s(g) \circ f_{\text{enc}}, \tag{2}$$

where $f_{\text{enc}} : \mathcal{X} \to \mathcal{Z}$ and $f_{\text{dec}} : \mathcal{Z} \to \mathcal{X}$ are encoder and decoder neural networks between the state space $\mathcal{X}$ and a latent space $\mathcal{Z} \subset \mathbb{R}^m$. $\pi_s : G \to \text{GL}(m)$ is a group representation such that $\pi_s(g) \in \mathbb{R}^{m \times m}$ is a matrix that can transform a latent vector $z \in \mathcal{Z}$ by matrix multiplication. GL is the general linear group. Under mild assumptions, authors in [40] prove the *universal approximation* of a nonlinear group action using the decomposed format in Equation (2). Consequently, we only need to investigate the equivariant relation on $\mathcal{Z}$, which can be written as follows:

$$\forall g \in G, \boldsymbol{z}(t_i) \in \mathcal{Z}, \ \ \pi_s(g)\boldsymbol{z}(t_{i+1}) = f_z(\pi_s(g)\boldsymbol{z}(t_i)). \tag{3}$$

where $f_z = f_{\text{enc}} \circ f \circ f_{\text{dec}}$ is the dynamic function in $\mathcal{Z}$. It's easy to verify that when Equations (2) and (3) hold and the encoder and decoder are well-trained (i.e., $f_{\text{dec}} \circ f_{\text{enc}} = I$, where $I$ is the identity matrix), the equivariant relation in Equation (1) also holds.

## 2.3 Time-Series Latent Space Construction

To construct the latent space for time-series measurements, we introduce Latent ODE [18] since it can explicitly model a continuous ODE process in $\mathcal{Z}$, capable of interpolating missing data at arbitrary times. In Latent ODE, ODE-RNN is the encoder ($f_{\text{enc}}$) to process $\{\boldsymbol{x}(t_i)\}_{i \in \mathcal{N}_x}$ and generate a latent vector $\boldsymbol{z}(t_0)$. The decoder ($f_{\text{dec}}$) employs a Neural ODE [16] to reconstruct the input state sequence and predict the future state. Mathematically, $f_{\text{enc}}$ approximates the posterior:

$$q_\phi(\boldsymbol{z}(t_0)|\{\boldsymbol{x}(t_i)\}_{i \in \mathcal{N}_x}) = \mathcal{N}(\mu_0, \Sigma_0), \ \mu_0, \Sigma_0 = f_{\text{enc}}(\{\boldsymbol{x}(t_i)\}_{i \in \mathcal{N}_x}), \tag{4}$$

where $\phi$ is the parameter set for $f_{\text{enc}}$ that approximates the mean and the variance of the posterior $q_\phi(\boldsymbol{z}(t_0)|\{\boldsymbol{x}(t_i)\}_{i \in \mathcal{N}_x})$. The encoding process is shown on the bottom left of Fig. 1. The decoder uses $\boldsymbol{z}(t_0)$ as the initial latent state and solves the ODE problem for inference:

$$\boldsymbol{z}(t_0) \sim p(\boldsymbol{z}(t_0)), \ \boldsymbol{z}(t_i) = \text{ODESolve}(h_\theta, \boldsymbol{z}(t_0), t_i), \ \boldsymbol{x}(t_i) \sim p_\theta(\boldsymbol{x}(t_i)|\boldsymbol{z}(t_i)), \tag{5}$$

where $\theta$ is the parameter set for $f_{\text{dec}}$, $p(\boldsymbol{z}(t_0))$ is a zero-mean Gaussian, $h : \mathcal{Z} \to \mathcal{Z}$ is a neural network to represent the derivative $\dot{\boldsymbol{z}}(t)$, and $p_\theta$ includes several MLP layers to approximate the conditional distribution. Thus, $f_{\text{dec}}$ consists of both $h_\theta$ and $p_\theta$. The overall training maximizes the evidence lower bound (ELBO): $\text{ELBO}(\phi, \theta) = \mathbb{E}_{\boldsymbol{z}(t_0) \sim q_\phi} \log p_\theta(\{\boldsymbol{x}(t_i)\}_{i \in \mathcal{N}_x}) - \text{KL}\big[q_\phi(\boldsymbol{z}(t_0)|\{\boldsymbol{x}(t_i)\}_{i \in \mathcal{N}_x})||p(\boldsymbol{z}(t_0))\big]$. For the deterministic scenario, we can also minimize the Mean Square Error (MSE). The bottom right of Fig. 1 shows the decoding process.

## 3 Method

In this section, we introduce a dynamic learning model that intrinsically preserves complex symmetries (Section 2.1) without symmetry discovery. By embedding geometric priors into the latent dynamics in Equation (3), we enforce physically grounded, symmetry-preserving structure with theoretical guarantees (Section 3.1). This foundation extends to a mixture model (Section 3.2) and generalizes to multi-scale temporal equivariance (Section 3.3).

## 3.1 Short-term Equivariance Preservation for A Single Latent Symmetry

We begin with the ideal case where the dynamics are governed by symmetry transformations of a single Lie group $G$. Then, we examine the functional class of dynamic models that have the potential to preserve equivariance. An expressive class is the dynamic function that can *commute with* the transformation in $G$. For example, the rotating dynamics in a 2D circle can commute with the rotational action in $\mathrm{SO}(2)$, and thus are $\mathrm{SO}(2)$-equivariant. SO is the special orthogonal group. In general, to study the commutative property, we demand the dynamic itself to be a Lie group action. Specifically, we consider the nonlinear Lie group action $\pi'_d(h, \cdot)$ such that:

$$\pi'_d(h, \cdot) = f_{\text{dec}} \circ \pi_d(h) \circ f_{\text{enc}}, \quad \boldsymbol{z}(t_{i+1}) = f_z(\boldsymbol{z}(t_i)) := \pi_d(h)\boldsymbol{z}(t_i), \tag{6}$$

where we use the subscript $d$ to respect the dynamics. $h \in H$ and $H$ is another Lie group that represents dynamics. By the universal approximation in Section 2.2, $\pi'_d(h, \cdot)$ can be decomposed, as shown in the first equality in Equation (6). Thus, we only need to focus on the property of the matrix transformation $\pi_d(h)$ operating in $\mathcal{Z}$. The following lemma illustrates how $\pi_d(h)$ can commute with $\pi_s(g)$ to preserve the equivariance.

**Lemma 1.** *Assume that $G$ and $H \subseteq G$ are Lie groups whose elements are defined in Equation (6) and Equation (3), respectively. Define the centralizer of $H$ in $G$ as: $C_G(h) := \{g \in G | \pi_s(g)\pi_d(h) = \pi_d(h)\pi_s(g)\}$. If $C_G(h)$ is nontrivial (i.e., contains elements other than the identity), then $\forall g \in C_G(h)$, the relation for $\pi_s(g)$ in Equation (3) is preserved.*

The proof is in Appendix A.1. The commutative property in the centralizer $C_G(h)$ defines an equivalent format for $G$-equivariance. In Lemma 1, the first condition ($h \in H \subseteq G$) is ideal by assuming a globally constant dynamic function and a single Lie group $G$. Hence, in Section 3.2, we design the learning model capable of capturing mixed and time-variant symmetries. The second condition ($C_G(h)$ is nontrivial) doesn't always hold. Thus, we establish the following conditions.

**Proposition 1** (Sufficient Conditions for Nontrivial Centralizer). *Let $H \subset \mathrm{Aff}(m)$ be an affine Lie subgroup whose elements are represented in homogeneous coordinates as affine transformations:*

$$\pi_d(h)\tilde{\boldsymbol{z}}(t) := \begin{bmatrix} A_h & \boldsymbol{b}_h \\ \boldsymbol{0} & 1 \end{bmatrix} \begin{bmatrix} \boldsymbol{z}(t) \\ 1 \end{bmatrix}, \tag{7}$$

*where $A_h \in \mathrm{GL}(m)$, $\boldsymbol{b}_h \in \mathbb{R}^m$, and we use $\tilde{\boldsymbol{z}}(t) = [\boldsymbol{z}(t), 1]^\top \in \mathbb{R}^{m+1}$ to denote the augmented latent vector in homogeneous coordinates to support affine transformations. Suppose that for $\forall g, h \in H$, it holds that $A_g A_h = A_h A_g$, $A_g \boldsymbol{b}_h = \boldsymbol{b}_h$ and $A_h \boldsymbol{b}_g = \boldsymbol{b}_g$. Then we have $H \subseteq C_G(h)$. In particular, $C_G(h)$ is nontrivial and contains at least all of elements in $H$.*

The proof is provided in Appendix A.2. In Equation (7), we adopt the upper-triangular affine matrix form because it offers a unified and compact representation for a broad class of Lie transformations, including *rotation*, *scaling*, and *translation*, and their higher-order multiplications. Subsequently, we show in Corollaries 1–3 that they satisfy the commutativity conditions in Proposition 1, thus having equivariance guarantees.

**Corollary 1** (Equivariance of Planar Rotation Transformation). *Let $u_1$ and $u_2$ be orthonormal vectors and $P := [u_1 \ u_2] \in \mathbb{R}^{m \times 2}$. Consider the planar rotation transformation: $\hat{\pi}^{rot} = \begin{bmatrix} I_m + P(R_\theta - I_2)P^\top & 0 \\ \boldsymbol{0} & 1 \end{bmatrix}$, where $R_\theta = \begin{bmatrix} \cos\theta & -\sin\theta \\ \sin\theta & \cos\theta \end{bmatrix} \in \mathrm{SO}(2)$, $I_m$ is the $m \times m$ identify matrix. Then, $\hat{\pi}^{rot}$ and its Lie subgroup $H \subset \mathrm{SO}(m)$ satisfy the conditions in Proposition 1.*

The restriction to planar rotation arises because general rotations are non-commutative. E.g., a die rotates along the x-axis and then z-axis, or along the z-axis and then x-axis, will cause different results. In contrast, planar rotations (fixed rotation axis) are commutative. The result is rigorously proved in Appendix A.3, meaning that the state should rotate along one axis at least locally to preserve rotation symmetry. If a rotation continuously changes the axis, symmetry is completely removed. Although $R_\theta$ is two-dimensional, the learnable projection matrix can lift the dimension to increase the expressiveness of $\hat{\pi}^{rot}$.

**Corollary 2** (Equivariance of Translation and Scaling Transformation). *Denote the translation dynamics: $\hat{\pi}^{tra} = \begin{bmatrix} I_m & \boldsymbol{v} \\ \boldsymbol{0} & 1 \end{bmatrix}$ and its Lie subgroup $H \subset \mathrm{E}(m)$, where $\mathrm{E}(m)$ is an Euclidean group. They satisfy the conditions in Proposition 1. Denote the scaling dynamics: $\hat{\pi}^{sca} = \begin{bmatrix} \mathrm{diag}(\boldsymbol{\gamma}) & \boldsymbol{0} \\ \boldsymbol{0} & 1 \end{bmatrix}$*

*and its Lie subgroup $H \subset \mathrm{Sim}(m)$, where $\mathrm{Sim}(m)$ is a similarity group. They satisfy the conditions in Proposition 1.*

**Corollary 3** (Equivariance of Second-Order Composed Transformations). *The second-order composition of the planar rotation, translation, and scaling transformations defined in Corollaries 1–2 satisfies the commutativity conditions in Proposition 1. Specifically, for any $\hat{\pi}^{(1)}, \hat{\pi}^{(2)} \in \{\hat{\pi}^{rot}, \hat{\pi}^{tra}, \hat{\pi}^{sca}\}$, we have that $\hat{\pi}^{(1)}\hat{\pi}^{(2)}$ and its Lie subgroup $H \subset \mathrm{Aff}(m)$ satisfy the conditions in Proposition 1.*

In corollary 1 to 3, we add $\hat{\ }$ to the matrix transformation to imply that it's a candidate dynamic approximation as we don't know the ground truth. The proof is provided in the Appendix A.3 to A.5.

## 3.2 Latent Mixture of Symmetries with Preserved Equivariance

Proposition 1 identifies a rich set of Lie groups that may appear in complex systems. In this section, we introduce our model to capture the mixture of diverse symmetries and maintain high representational power. Fig. 1 demonstrates the main architecture. The core is a Mixture-of-Experts (MoE) model, which employs a gating mechanism [49] to automatically select the appropriate Lie group transformations. In addition, we make the MoS module time-variant for time-dependent symmetries. Thanks to the model's extensibility, new Lie group experts that are not characterized by Proposition 1 can be flexibly incorporated. We leave such extensions for future work.

Specifically, let $h \in \mathbb{R}^K$ be the output of a gating neural network, activated by a Softmax function. Treating each symmetry transformation as an expert, the $k^{th}$ entry $h[k]$ provides the weight for the expert. Then, Latent MoS linearly mixes the latent flows $\tilde{z}_k(t_{i+1})$ such that:

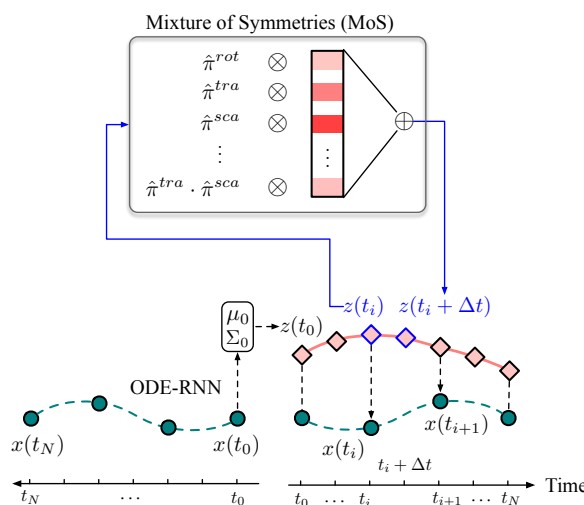

Figure 1: The Latent MoS framework. Compared to Latent ODE, Latent MoS *structures* the latent dynamics by using an MoE to select symmetries. Different colors in the MoS box imply different gate weights.

$$\tilde{z}(t_{i+1}) = \sum_{k=1}^{K} h[k] \cdot \tilde{z}_k(t_{i+1}) := \sum_{k=1}^{K} h[k] \cdot \hat{\pi}_k \cdot \tilde{z}(t_i), \qquad (8)$$

where $\hat{\pi}_k$ is the $k^{th}$ expert, parameterized to represent a Lie group action with a geometric structure from Proposition 1. We adopt a linear mixture because real-world systems often exhibit multiple latent symmetry flows acting simultaneously (e.g., damping and oscillation), which cannot be represented by a single Lie group action. The linear mix allows the gating network to combine these expert transformations into expressive latent dynamics, while still preserving the interpretability of each individual symmetry component.

In Equation (8), $h$ and $\hat{\pi}_k$ are made of time-dependent. Specifically, let the time interval $[t_0, t_N]$ be evenly divided into $L$ disjoint subintervals:

$$[t_0, t_N] = \bigcup_{l=0}^{L-1} [t^{(l)}, t^{(l+1)}), \quad \text{where} \quad t^{(l)} := t_0 + l \cdot \Delta T, \ \Delta T = \frac{t_N - t_0}{L}. \qquad (9)$$

Then, for each subinterval, $\hat{\pi}_k$ and $h$ are piecewise constant and locally dependent on both the initial time $t^{(l)}$ and the latent state $\tilde{z}(t^{(l)})$, i.e., $h := h(t^{(l)}, z(t^{(l)}))$ and $\hat{\pi}_k := \hat{\pi}_k(t^{(l)}, z(t^{(l)}))$. The specific designs for $\hat{\pi}_k$ are shown below. We eliminate the superscript $(l)$ for simplicity.

**Rotational symmetry**. The group action $\hat{\pi}^{rot}(t, z(t)) \in \mathrm{SE}(m)$, where $\mathrm{SE}(m)$ is a special Euclidean group. To parameterize $\hat{\pi}^{rot}(t, z(t))$, by Corollary 1, we first learn a projection matrix $\hat{P}(t, z(t)) \in \mathbb{R}^{m \times 2}$, whose columns form an orthonormal basis of a 2D subspace in $\mathbb{R}^m$. Specifically, a neural network outputs an unconstrained matrix $\hat{V}(t, z(t)) \in \mathbb{R}^{m \times 2}$, which is then orthonormalized via a QR decomposition: $\hat{V}(t, z(t)) = \hat{Q}(t, z(t))\hat{R}(t, z(t))$, and we define $\hat{P}(t, z(t)) := \hat{Q}(t, z(t))$. We also parameterize a planar rotation matrix $R_\theta \in \mathrm{SO}(2)$ using a learnable scalar angle $\theta(t, z(t))$. The

overall transformation is then defined as: $\hat{\pi}^{rot}(t, \boldsymbol{z}(t)) := \begin{bmatrix} I_m + \hat{P}(R_\theta - I_2)\hat{P}^\top & \mathbf{0} \\ \mathbf{0}^\top & 1 \end{bmatrix} \in \mathrm{SE}(m)$,

which applies a learned planar rotation within a dynamically selected 2D subspace while leaving the rest of the space invariant and preserving the affine structure.

**Translational symmetry.** The action $\hat{\pi}^{tra}(t, \boldsymbol{z}(t)) \in \mathrm{E}(m)$, where $\mathrm{E}(m)$ is an Euclidean group. Given a learnable velocity vector $\boldsymbol{v}(t, \boldsymbol{z}(t)) \in \mathbb{R}^m$, we define: $\hat{\pi}^{tra}(t, \boldsymbol{z}(t)) = \begin{bmatrix} I & \boldsymbol{v}(t, \boldsymbol{z}(t)) \\ \mathbf{0} & 1 \end{bmatrix}$.

**Scaling symmetry.** The group action $\hat{\pi}^{sca}(t, \boldsymbol{z}(t)) \in \mathrm{Sim}(m)$, where $\mathrm{Sim}(m)$ is a similarity group. For a ratio vector $\boldsymbol{\gamma}(t, \boldsymbol{z}(t)) \in \mathbb{R}^m$, we have $\hat{\pi}^{sca}(t, \boldsymbol{z}(t)) = \begin{bmatrix} \mathrm{diag}(\boldsymbol{\gamma}(t, \boldsymbol{z}(t))) & \mathbf{0} \\ \mathbf{0} & 1 \end{bmatrix}$.

**Second-order multiplications.** To capture coupled symmetry effects, we consider the multiplication of multiple Lie group actions. For example, multiplying a translational action $\hat{\pi}^{tra}(t, \boldsymbol{z}(t))$ with a scaling action $\hat{\pi}^{sca}(t, \boldsymbol{z}(t))$ yields a new transformation $\hat{\pi}^{com}(t, \boldsymbol{z}(t)) = \hat{\pi}^{\mathrm{tra}}(t, \boldsymbol{z}(t)) \cdot \hat{\pi}^{\mathrm{sca}}(t, \boldsymbol{z}(t))$. More generally, the multiplication of Lie group actions remains a valid Lie group action when defined over a closed subgroup (e.g., $\mathrm{SE}(m)$, $\mathrm{E}(m)$, $\mathrm{Sim}(m)$, etc.).

Fig. 1 illustrates how the gating mechanism enables a mixture of symmetry transformations to govern the latent dynamics. Compared to the Latent ODE model introduced in Section 2.3, our Latent Mixture of Symmetries (Latent MoS) replaces the unstructured ODE update step (i.e., ODESolve in Equation (5)) with a structured, symmetry-preserving formulation as given in Equation (8). According to Proposition 1, each latent flow $\tilde{\boldsymbol{z}}_k(t)$ in Equation (8) preserves an underlying equivariance, which facilitates more sample-efficient learning.

**Discrete vs. continuous updates.** Equation (8) adopts a discrete update formulation based on Lie group actions. However, our framework also admits a continuous-time integral by computing the matrix logarithm of the group action to obtain its infinitesimal generator. Specifically, for each expert, we define $\hat{\xi}_k(t, \boldsymbol{z}(t)) := \log(\hat{\pi}_k(t, \boldsymbol{z}(t)))$, where $\hat{\xi}_k \in \mathfrak{g}$ is the Lie algebra element corresponding to $\hat{\pi}_k \in G$. This enables continuous integration of the latent trajectory via the Lie algebra. Specifically, for the $k^{th}$ flow, the evolution of $\tilde{\boldsymbol{z}}(t)$ is governed by the differential equation $\frac{d}{dt}\tilde{\boldsymbol{z}}_k(t) = \hat{\xi}_k(t, \boldsymbol{z}(t)) \cdot \tilde{\boldsymbol{z}}(t)$, whose solution over a small interval $[t_i, t_i + \Delta t]$ is given by the matrix integral: $\tilde{\boldsymbol{z}}_k(t_i + \Delta t) = \exp\left(\int_{t_i}^{t_i+\Delta t} \hat{\xi}_k(s, \boldsymbol{z}(s))\, ds\right) \cdot \tilde{\boldsymbol{z}}(t_i)$. While this formulation enables interpolation and handling of irregular time intervals, computing the integral and exponential map can be computationally expensive. By the first-order approximation of Lie group flows [50], we can approximate the matrix integral as $\tilde{\boldsymbol{z}}_k(t_i + \Delta t) \approx \exp\left(\Delta t \cdot \hat{\xi}_k(t_i, \boldsymbol{z}(t_i))\right) \cdot \tilde{\boldsymbol{z}}(t_i)$. With a slight abuse of notation, we then define the group action $\hat{\pi}_k(t^{(i)}, \boldsymbol{z}(t^{(i)})) := \exp\left(\Delta t \cdot \hat{\xi}_k(t^{(i)}, \boldsymbol{z}(t^{(i)}))\right)$, where $t_i \in [t^{(i)}, t^{(i+1)})$. This leads to the the discrete $\Delta t$ update:

$$\tilde{\boldsymbol{z}}(t_i + \Delta t) := \sum_{k=1}^{K} h(t^{(i)}, \boldsymbol{z}(t^{(i)}))[k] \cdot \hat{\pi}_k(t^{(i)}, \boldsymbol{z}(t^{(i)})) \cdot \tilde{\boldsymbol{z}}(t_i). \tag{10}$$

Combing this equation with the Equation (8), we can obtain a high-resolution latent sequence $\tilde{\boldsymbol{z}}_{0:N}(\Delta t) = (\tilde{\boldsymbol{z}}(t_0), \tilde{\boldsymbol{z}}(t_0 + \Delta t), \tilde{\boldsymbol{z}}(t_0 + 2\Delta t), \cdots, \tilde{\boldsymbol{z}}(t_N - \Delta t), \tilde{\boldsymbol{z}}(t_N))$, which can be treated as features for interpolation and extrapolation, as shown in Equation (5). The bottom right part of Fig. 1 demonstrates the process: the green observations are sparse, but the pink latent vectors are dense.

### 3.3 Multi-scale Latent MoS for Short- and Long-term Equivariance

The transformation $\hat{\pi}_k(t^{(i)}, \boldsymbol{z}(t^{(i)}))$, defined only within the local interval $[t^{(l)}, t^{(l+1)})$, may fail to capture long-term equivariant relations. Specifically, there may exist a symmetry transformation $\pi_s(g)$ that maps $\boldsymbol{z}(t_i)$ to a point outside the current interval to preserve the dynamics, but it will be missed by locally defined models. To address this, we propose a simple hierarchical structure. Intuitively, we consider different durations for a constant dynamic function, illustrated in Algorithm 1. $L^i > L^j$ if level $j$ corresponds to a higher and coarser resolution level.

---

**Algorithm 1** Multi-level Latent Sequence Generation

---

**Require:** Initial full latent state $\tilde{\boldsymbol{z}}(t_0) \in \mathbb{R}^m$, time step $\Delta t$, number of subintervals at different levels $\{L^{(1)}, L^{(2)}, \ldots, L^{(S)}\}$.

1: Partition $\tilde{\boldsymbol{z}}(t_0)$ into $[\tilde{\boldsymbol{z}}^{(1)}(t_0), \ldots, \tilde{\boldsymbol{z}}^{(S)}(t_0)]$, where each $\tilde{\boldsymbol{z}}^{(s)}(t_0) \in \mathbb{R}^{m/S}$.
2: **for** each level $s = 1$ to $S$ **do**
3:     Set $L \leftarrow L^{(s)}$. Use Equation (9) to compute the subintervals $\{[t^{(l)}, t^{(l+1)})\}_{l=0}^{L-1}$
4:     **for** each subinterval $[t^{(l)}, t^{(l+1)})$ **do**.
5:         Generate MoS$^{(s)}$: $\{\hat{\pi}_k(t^{(l)}, \tilde{\boldsymbol{z}}^{(s)}(t^{(l)}))\}_{k=1}^K$ and gates $h^{(s)}(t^{(l)}, \tilde{\boldsymbol{z}}^{(s)}(t^{(l)}))$.
6:     **end for**
7:     Use Equation (10) to gain a trajectory $\tilde{\boldsymbol{z}}_{0:N}^{(s)}(\Delta t) = \Big(\tilde{\boldsymbol{z}}^{(s)}(t_0), \tilde{\boldsymbol{z}}^{(s)}(t_0 + \Delta t), \ldots, \tilde{\boldsymbol{z}}^{(s)}(t_N)\Big)$.
8: **end for**
9: **return** $\tilde{\boldsymbol{z}}_{0:N}(\Delta t) = $ concat $\Big(\tilde{\boldsymbol{z}}_{0:N}^{(1)}(\Delta t), \ldots, \tilde{\boldsymbol{z}}_{0:N}^{(S)}(\Delta t)\Big)$ along the feature dimension.

---

## 4 Experiments

We test the following datasets, fully described in Appendix B. (1) **Complex Nonlinear ODE Systems**. Similar to [37], we employ spiral datasets, Glycolytic oscillators (biochemical system), and Lotka-Volterra systems (the interaction between a predator and a prey population). (2) **Residential Electricity Consumption**. We gather real-world electricity data, publicly available at [51–53]. (3) **Photovoltaic Solar Energy**. We introduce a publicly available Photovoltaic (PV) dataset [54] for solar power generations. (4) **Power System Event Measurements**. This is a 10-dimensional dataset (see Appendix B) produced by a high-order and time-dependent ODE system [55–57]. (5) **Air Quality System**. UCI Repository provides measurements of metal oxide chemical sensors in an air quality monitoring system [58]. (6) **Electrocardiogram (ECG) signals**. Recorded electrical signals from a patient's heart in the UCR Time Series Archive [59]. These test systems are representative, encompassing nonlinear and high-dimensional dynamics, as well as real-world applications in power systems, weather systems, and biomedical domains, all of which exhibit disturbances and noise. For these datasets, the interpolation and extrapolation tasks are illustrated in Appendix B.1. To evaluate these systems, we adopt a range of state-of-the-art methods based on Neural ODEs and Transformer, as detailed in Appendix C. Model architectures and training time comparisons are provided in Appendix D and Appendix E.3, respectively. For each method, we run 10 times and report the average errors.

### 4.1 Effective Symmetry Preservation in the Latent Space $\mathcal{Z}$

First, we validate whether Latent MoS can correctly preserve the symmetry. We test the model in the ODE systems. As shown in the left part of Fig. 2, the true data are governed by linear or nonlinear rotational and slight scaling symmetry. Then, we set the data drop rate to be $90\%$. We utilize Principal Component Analysis to reduce the 15-dimensional latent vector ($m = 15$) in $\mathcal{Z}$ to 2D for visualizations in the middle and right parts of Fig. 2. The result suggests that even with limited input data, latent MoS captures the correct and highly interpretable latent geometry, which can be used for the geometry-based analyses in dynamical systems. However, Latent ODE will produce unstructured latent trajectories that risk overfitting. The predicted curve visualizations are provided in Appendix E.2.

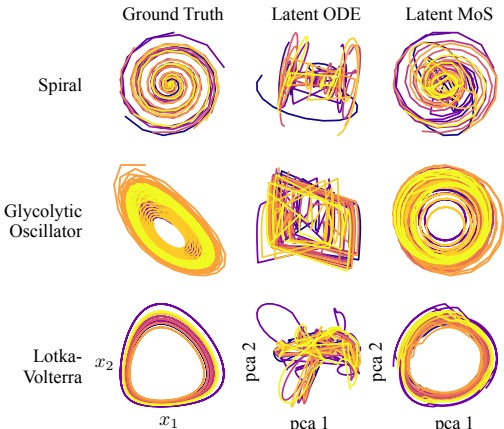

Figure 2: Raw data (left) and latent trajectories for Latent ODE (middle) and Latent MoS (right).

In the Glycolytic system, even when all symmetry experts are included, the model achieves a low MSE of 0.0033. Importantly, the gating weights are highly concentrated: the dominant expert, i.e., a second-order transformation (rotation × scaling), receives a weight of 0.97, while the next highest (scaling × rotation) receives 0.025, and the remaining experts collectively contribute less

Table 1: Test Mean Squared Error (MSE) ($\times 10^{-2}$) for extrapolation tasks under varying drop rates.

| Data | Drop | EqSINDy | EqGP | RNN-$\Delta_t$ | ODE-RNN | NCDE | Contiformer | Informer | Autoformer | Latent ODE | **Latent MoS** |
|---|---|---|---|---|---|---|---|---|---|---|---|
| Spiral | 90% | 1.79 | 2.01 | 1.01 | 1.16 | 1.34 | 1.07 | 1.21 | 1.13 | 1.04 | **0.62** |
| | 60% | 0.88 | 0.96 | 0.49 | 0.98 | 1.31 | 1.09 | 1.09 | 0.89 | 0.99 | **0.54** |
| | 30% | 0.45 | 0.78 | 0.33 | 0.95 | 1.11 | 1.09 | 1.10 | 0.93 | 0.96 | **0.33** |
| Glycolytic | 90% | 10.23 | 13.12 | 0.52 | 13.44 | 12.11 | 13.57 | 12.92 | 2.47 | 9.88 | **0.32** |
| | 60% | 8.81 | 13.06 | 0.20 | 13.29 | 11.23 | 7.54 | 0.77 | 0.49 | 3.88 | **0.15** |
| | 30% | 8.97 | 10.98 | 0.33 | 13.28 | 11.10 | 3.51 | **0.06** | 0.16 | 2.67 | **0.06** |
| Lotka | 90% | 8.70 | 10.66 | 0.81 | 5.39 | 13.49 | 6.30 | 14.50 | 6.55 | **0.43** | 0.61 |
| | 60% | 5.34 | 11.23 | 0.75 | 5.35 | 12.28 | 4.25 | 14.51 | 5.48 | 0.18 | **0.13** |
| | 30% | 3.98 | 10.75 | 0.35 | 5.32 | 11.10 | 2.49 | 14.50 | 1.80 | 0.09 | **0.05** |
| Load | 90% | 7.88 | 7.67 | 8.69 | 8.01 | 18.39 | 10.57 | 8.70 | 18.13 | 5.18 | **3.25** |
| | 60% | 4.98 | 4.27 | 5.75 | 6.39 | 15.19 | 5.49 | 8.39 | 9.81 | 3.61 | **2.42** |
| | 30% | 3.14 | 3.06 | 5.10 | 5.93 | 11.19 | 5.14 | 7.89 | 6.27 | 2.96 | **1.33** |
| Solar | 90% | 20.13 | 22.98 | 17.16 | 10.53 | 25.33 | 19.82 | 24.50 | 20.44 | 11.26 | **8.83** |
| | 60% | 21.25 | 20.41 | 12.77 | 7.62 | 24.19 | 17.43 | 15.58 | 15.23 | 8.17 | **5.01** |
| | 30% | 16.82 | 18.37 | 11.38 | 7.36 | 19.84 | 15.53 | 15.31 | 16.08 | 7.28 | **3.48** |
| Power event | 90% | 7.72 | 8.21 | 8.04 | 9.43 | 10.45 | 8.09 | 10.05 | 8.65 | 8.02 | **5.32** |
| | 60% | 6.43 | 8.89 | 7.95 | 8.84 | 9.08 | 7.23 | 8.17 | 7.85 | 6.98 | **3.75** |
| | 30% | 5.89 | 5.62 | 7.69 | 8.41 | 8.98 | 6.74 | 7.22 | 7.58 | 6.78 | **2.53** |
| Air quality | 90% | 10.09 | 9.22 | 9.38 | 10.50 | 11.20 | 8.44 | 8.28 | 10.56 | 9.55 | **5.90** |
| | 60% | 10.07 | 8.71 | 8.91 | 8.86 | 10.59 | 7.11 | 10.38 | 12.21 | 7.89 | **4.10** |
| | 30% | 9.13 | 8.98 | 7.95 | 6.91 | 9.05 | 6.89 | 7.99 | 9.44 | 6.40 | **2.69** |
| ECG | 90% | 5.98 | 6.92 | 3.48 | 4.33 | 5.33 | 3.49 | 4.08 | 3.91 | 3.04 | **1.08** |
| | 60% | 5.13 | 5.12 | 2.02 | 3.23 | 4.59 | 2.13 | 2.11 | 4.10 | 1.03 | **0.91** |
| | 30% | 4.29 | 3.77 | 1.84 | 2.70 | 4.19 | 1.04 | 1.49 | 3.48 | 1.02 | **0.62** |

than 0.005. This concentration indicates that the model identifies and relies on the correct symmetry transformation.

## 4.2 Overall Evaluations on Diverse Datasets with Different Data Drop Rates

We evaluate the model performance under varying drop rates: 90%, 60%, and 30%. The complete results are shown in Fig. 3 (see also Table 3 in Appendix E.1) for interpolation tasks and Table 1 for extrapolation tasks. In general, Contiformer, Latent ODE, and the proposed Latent MoS all achieve strong performance across both continuous ODE-based systems and real-world datasets with high disturbances. This is attributed to their shared ability to reconstruct latent trajectories during decoding, effectively capturing the underlying system dynamics. Among them, Latent MoS leverages geometric priors to structure the latent trajectories (see Fig. 2). For instance, while Contiformer and Latent ODE struggle with high-frequency Glycolytic oscillations, Latent MoS successfully encodes the data into a latent space $\mathcal{Z}$ where linear Lie group action-based dynamics can be more easily estimated. Across all datasets, Latent MoS achieves relative improvements ranging from 15% to 97%. In contrast, Informer and Autoformer only perform well under low drop rates, as they rely on pre-interpolation and lack the capacity to model the underlying continuous dynamics directly.

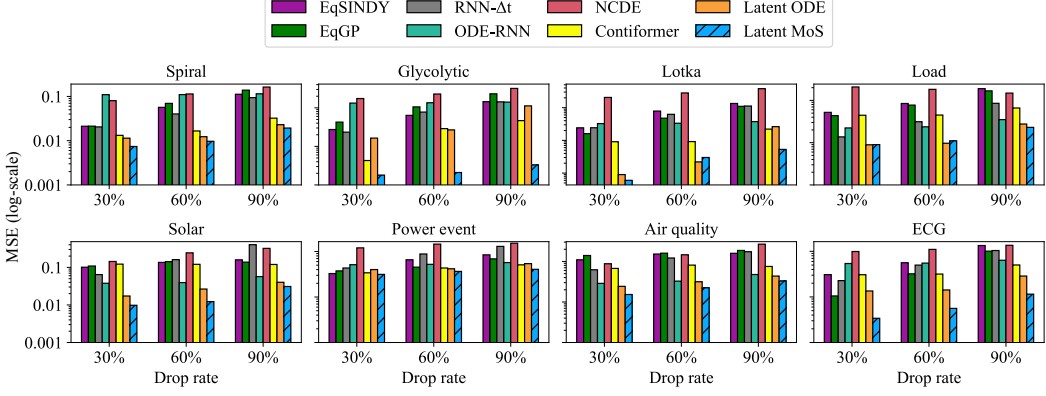

Figure 3: Test MSE in log-scales for interpolation tasks. Table 3 in Appendix E.1 shows values.

## 4.3 Sensitivity Analysis on Noise, Gating Mechanisms, and Latent Dimensionality

We use the Glycolytic system as an example for sensitivity analysis under a 90% data drop rate. Similar trends are observed across other systems. Fig. 4 shows the interpolation test set MSE under

various conditions. First, we introduce Gaussian noise with increasing standard deviations. The MSE for both Latent ODE and Latent MoS rises gradually, reflecting the impact of noise. This demonstrates that they are robust to noise perturbations and capable of approximating the underlying average trend. Second, we vary the latent dimensionality $m$. Latent MoS exhibits a sharp drop in MSE when $m > 5$. This suggests that a higher $m$ provides sufficient capacity for approximating the nonlinear Lie group actions described in Equation (2), while still preserving linear operations within $\mathcal{Z}$. In contrast, Latent ODE lacks structured inductive biases in its latent space, and its performance remains relatively unaffected by increased dimensionality. Third, we examine the effect of the gating mechanism by allowing the top $K_0 \leq K$ gates with the highest scores to open and vary $K_0$. The error remains when $K_0 \in [1, 4]$ because Glycolytic system contains rotational and scaling symmetries. However, when $K_0$ becomes large, Latent MoS tends to overfit. Empirically, we find that setting $K_0 = 4$ yields strong performance, though further improvements can be achieved through hyperparameter tuning.

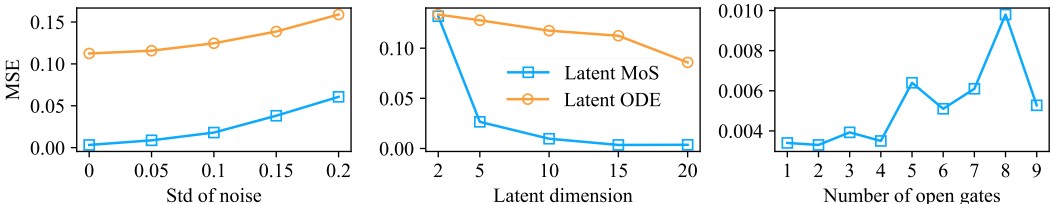

Figure 4: Results in sensitivity analysis for Glycolytic systems under $90\%$ data drops.

**Scalability in high-dimensional systems**. Consistent with empirical findings in Latent ODEs [18], we observe that Latent MoS achieves strong performance on the 10-dimensional power event dataset by setting $m = 30$. We hypothesize that this efficiency stems from underlying physical correlations.

## 4.4 Ablation Study

Using the same setting as in Section 4.3, we conduct an ablation study. As shown in Fig. 5, the presence of sparse gates is crucial for selecting relevant symmetries in our MoS framework. Furthermore, in the Glycolytic system, removing scaling or rotational symmetries significantly increases the MSE. Interestingly, performance improves when unnecessary symmetries, such as translational or certain second-order components, are excluded. When the translational expert is removed (which is unnecessary for Glycolytic dynamics), the MSE decreases to 0.001 and the gating weight for rotation $\times$ scaling increases to 0.99.

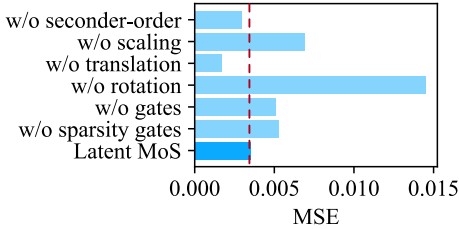

Figure 5: Results in ablation studies.

This demonstrates that while the gating works well, a simplified expert set can enhance efficiency and robustness. To achieve the best performance, we can perform a validation-based procedure to progressively remove unnecessary experts and compare the resulting MSE. Importantly, the gating mechanism remains valuable in this process: it helps identify the dominant expert(s) within a candidate subset, effectively narrowing down the search space.

## 4.5 Dynamic Learning for Control

We introduce an optimal frequency control problem in power systems. The task is to stabilize the generator's frequency. Past methods [60] assume an idealistic second-order ODE model, i.e., swing equations, to model frequency dynamics. However, real-world dynamics are higher-order with unknown physical parameters. Hence, we utilize latent MoS to learn it. In general, we compare model-free proximal policy optimization (PPO) [61], model-based policy optimization (MBPO) with a Gaussian process (MBPO-GP) [62], MBPO with Latent MoS (MBPO-MoS), and MBPO with the swing equation (MBPO-SE). By the end time, our MBPO-MoS stabilizes near zero with a deviation $-0.024$, the lowest among all methods. In contrast, PPO remains highly unstable ($-0.928$), MBPO-GP yields $0.100$, and MBPO-SE yields $-0.115$. The general cost (integral of the Euclidean norm of frequency deviations) in the control process is $0.081$ (MBPO-MoS), $0.163$ (MBPO-SE), $0.257$ (MBPO-GP), and $0.51$ (PPO). Our learned dynamics in Latent MoS can much better predict the future frequency, hence largely improving control performance.

# 5 Related Work

**Dynamic Modeling Against Data Scarcity**. Existing approaches to dynamic modeling under data scarcity fall into four categories. The first focuses on interpolation: model-based methods (e.g., multidimensional interpolation [63], physical estimations [64]) assume explicit system behavior, while optimization-based techniques (e.g., compressed sensing [8], matrix completion [15]) rely on low-rank or sparsity assumptions to ensure tractability. Data-driven models from signal processing and machine learning [65, 12] can adapt to complex patterns but often neglect domain-specific structures. The second category incorporates physical knowledge through governing equations [11, 12, 66], enabling convergence even from sparse data. The third uses Bayesian inference to inject priors and quantify uncertainty [13–15], though performance depends heavily on prior quality. The fourth employs neural ODE solvers [16], including Latent ODEs [18] and Neural CDEs [17], which model latent continuous-time flows decodable to system states but require high-resolution data, as ODE integral errors accumulate [67].

**Symmetry-based Learning**. Enforcing symmetry priors in equivariant neural networks improves generalization and sample efficiency for applications like image processing [27–30] and RL [31–33]. However, they typically demand prior knowledge to handcraft symmetry transformations. To discover unknown symmetry knowledge, pioneer work involves a restrictive search space [68–70, 42, 38], e.g., a discrete symmetry group [39]. Recent work extends this study to generate and approximate different kinds of symmetries [40, 41, 37]. In particular, the framework based on Generative Adversarial Networks (GANs) [40, 41, 37] is highly generalizable to finding different symmetries in the latent space, grounded on the theory of Lie algebra. Nevertheless, GAN is known to have training instability [71], and their methods may deliver fallacious, trivial, or unclear solutions [41].

**Symmetry-breaking Learning**. Existing approaches that incorporate symmetry breaking in machine learning can be categorized into three main groups. (1) Learnable or non-stationary weights relax strict equivariance by allowing model parameters to adapt locally [44–46]. (2) Soft regularization encourages but does not strictly enforce equivariance, typically through auxiliary loss terms [43]. (3) Symmetry-breaking sets provide a constructive strategy by introducing minimal auxiliary inputs to selectively break output symmetry [72]. Our model falls into the first category, employing locally adaptive, learnable weights to capture approximate symmetries.

# 6 Conclusion, Limitation, and Future Work

We present Latent MoS, a sample-efficient framework for learning dynamical systems that explicitly preserves equivariance through structured latent transformations. Motivated by the observation that real-world systems often exhibit symmetry breaking, where strict group invariances hold only approximately or locally, Latent MoS captures these effects through mixtures of symmetry groups and adaptable latent flows. Grounded in Lie group theory, our approach provides a physically interpretable and effective means of modeling system dynamics under symmetry constraints. Although the set of predefined candidate transformations does not cover the full space of possible Lie group representations, empirical results demonstrate consistent improvements across a wide range of physical systems. The modular design of Latent MoS further enables natural extensions to incorporate additional or more complex symmetry transformations.

Latent MoS is primarily suited for systems that exhibit at least local symmetries. While systems entirely lacking symmetry, such as highly chaotic or purely stochastic processes (e.g., unconstrained turbulence), may fall outside its intended scope, such cases are rare in practice. Most real-world physical, biological, and engineered systems exhibit local or approximate symmetries due to conservation laws, structural regularities, or recurring patterns.

In future work, we plan to extend Latent MoS in two directions. First, we will develop automated mechanisms to refine and constrain the candidate set of symmetries, using validation-based selection or search strategies, so that the model can achieve the best performance even when the true symmetries are uncertain. Second, we aim to explore physically meaningful symmetry structures for broader geometric analyses in dynamical systems, including stability, safety, and contraction properties.

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

## Impact Statement

This paper presents work whose goal is to advance the field of Machine Learning. There are many potential societal consequences of our work, none which we feel must be specifically highlighted here.

# A    Detailed Proofs

## A.1    Proof of Lemma 1

**Lemma 1.** *Assume that $G$ and $H \subseteq G$ are Lie groups whose elements are defined in Equation (6) and Equation (3), respectively. Define the centralizer of $H$ in $G$ as: $C_G(h) := \{g \in G | \pi_s(g)\pi_d(h) = \pi_d(h)\pi_s(g)\}$. If $C_G(h)$ is nontrivial (i.e., it contains elements other than the identity), then $\forall g \in C_G(h)$, the relation for $\pi_s(g)$ in Equation (3) is preserved.*

*Proof.* For $\forall g \in C_G(h)$, we have

$$\pi_s(g)\boldsymbol{z}(t_{i+1}) = \pi_s(g)\pi_d(h)\boldsymbol{z}(t_i) \tag{11}$$

$$= \pi_d(h)\pi_s(g)\boldsymbol{z}(t_i) \tag{12}$$

$$= f_z(\pi_s(g)\boldsymbol{z}(t_i)), \tag{13}$$

where the first equality follows from Equation (6), the second by the definition of the centralizer, and the third again by Equation (6). $\qquad\square$

## A.2    Proof of Proposition 1

**Proposition 1** (Sufficient Conditions for Nontrivial Centralizer). *Let $H \subset \mathrm{Aff}(m)$ be a Lie subgroup whose elements are represented in homogeneous coordinates as affine transformations:*

$$\pi_d(h)\tilde{\boldsymbol{z}}(t) = \pi_d(h)(\begin{bmatrix} \boldsymbol{z}(t) \\ 1 \end{bmatrix}) = \begin{bmatrix} A_h & \boldsymbol{b}_h \\ \boldsymbol{0} & 1 \end{bmatrix}\begin{bmatrix} \boldsymbol{z}(t) \\ 1 \end{bmatrix}, \tag{14}$$

*where $A_h \in \mathrm{GL}(m)$, $\boldsymbol{b}_h \in \mathbb{R}^m$, and we use $\tilde{\boldsymbol{z}}(t) = [\boldsymbol{z}(t), 1]^\top \in \mathbb{R}^{m+1}$ to denote the augmented latent vector in homogeneous coordinates to support affine transformations. Suppose that for $\forall g, h \in H$, it holds that $A_g A_h = A_h A_g$ and $A_g \boldsymbol{b}_h = \boldsymbol{b}_h$. Then we have $H \subseteq C_G(h)$. In particular, $C_G(h)$ is nontrivial and contains at least all of elements in $H$.*

*Proof.* We notice that

$$\pi_d(g)\,\pi_d(h) = \begin{bmatrix} A_g & \boldsymbol{b}_g \\ \boldsymbol{0} & 1 \end{bmatrix}\begin{bmatrix} A_h & \boldsymbol{b}_h \\ \boldsymbol{0} & 1 \end{bmatrix} = \begin{bmatrix} A_g A_h & A_g \boldsymbol{b}_h + \boldsymbol{b}_g \\ \boldsymbol{0} & 1 \end{bmatrix}, \tag{15}$$

and

$$\pi_d(h)\,\pi_d(g) = \begin{bmatrix} A_h & \boldsymbol{b}_h \\ \boldsymbol{0} & 1 \end{bmatrix}\begin{bmatrix} A_g & \boldsymbol{b}_g \\ \boldsymbol{0} & 1 \end{bmatrix} = \begin{bmatrix} A_h A_g & A_h \boldsymbol{b}_g + \boldsymbol{b}_h \\ \boldsymbol{0} & 1 \end{bmatrix}. \tag{16}$$

Since for $\forall g, h \in H$, $A_g A_h = A_h A_g$ and $A_g \boldsymbol{b}_h = \boldsymbol{b}_h$, then we have $\pi_d(g)\,\pi_d(h) = \pi_d(h)\,\pi_d(g)$. It shows that every pair of elements $g, h \in H$ commute under $\pi_d$. Thus, $g \in C_G(h), \forall g \in H$. Thus, the centralizer is nontrivial and contains at least $H$. $\qquad\square$

### A.3 Proof of Corollary 1

**Corollary 1** (Equivariance of Planar Rotation Transformation). *Let $u_1$ and $u_2$ be orthonormal vectors and $P := [u_1 \; u_2] \in \mathbb{R}^{m \times 2}$. Consider the planar rotation transformation: $\hat{\pi}^{rot} = \begin{bmatrix} I_m + P(R_\theta - I_2)P^\top & 0 \\ \mathbf{0} & 1 \end{bmatrix}$, where $R_\theta = \begin{bmatrix} \cos\theta & -\sin\theta \\ \sin\theta & \cos\theta \end{bmatrix} \in \mathrm{SO}(2)$, $I_m$ is the $m \times m$ identify matrix. Then, $\hat{\pi}^{rot}$ and its Lie group $H \subset \mathrm{SO}(m)$ satisfies the conditions in Proposition 1.*

*Proof.* For the rotation matrix $Q_\theta = I_m + P(R_\theta - I_2)P^\top$, we have

$$
\begin{aligned}
Q_\theta Q_\gamma &= (I_m + P(R_\theta - I_2)P^\top)(I_m + P(R_\gamma - I_2)P^\top) \\
&= I_m + P(R_\theta - I_2)P^\top P(R_\gamma - I_2)P^\top + P(R_\theta - I_2)P^\top + P(R_\gamma - I_2)P^\top \\
&= I_m + P(R_\theta - I_2)(R_\gamma - I_2)P^\top + P(R_\theta - I_2)P^\top + P(R_\gamma - I_2)P^\top
\end{aligned}
$$

and

$$
\begin{aligned}
Q_\gamma Q_\theta &= (I_m + P(R_\gamma - I_2)P^\top)(I_m + P(R_\theta - I_2)P^\top) \\
&= I_m + P(R_\gamma - I_2)P^\top P(R_\theta - I_2)P^\top + P(R_\gamma - I_2)P^\top + P(R_\theta - I_2)P^\top \\
&= I_m + P(R_\gamma - I_2)(R_\theta - I_2)P^\top + P(R_\gamma - I_2)P^\top + P(R_\theta - I_2)P^\top.
\end{aligned}
$$

where the equality holds due to $P^\top P = I_2$. Since we have

$$
\begin{aligned}
R_\theta R_\gamma &= \begin{bmatrix} \cos\theta & -\sin\theta \\ \sin\theta & \cos\theta \end{bmatrix} \begin{bmatrix} \cos\gamma & -\sin\gamma \\ \sin\gamma & \cos\gamma \end{bmatrix} = \begin{bmatrix} \cos\theta\cos\gamma - \sin\theta\sin\gamma & -\cos\theta\sin\gamma - \sin\theta\cos\gamma \\ \sin\theta\cos\gamma + \cos\theta\sin\gamma & -\sin\theta\sin\gamma + \cos\theta\cos\gamma \end{bmatrix} \\
&= \begin{bmatrix} \cos(\theta+\gamma) & -\sin(\theta+\gamma) \\ \sin(\theta+\gamma) & \cos(\theta+\gamma) \end{bmatrix} = R_{\theta+\gamma} = R_\gamma R_\theta,
\end{aligned}
$$

we also have $Q_\theta Q_\gamma = Q_\gamma Q_\theta$. $\qquad\square$

### A.4 Proof of Corollary 2

**Corollary 2** (Equivariance of Translation and Scaling Transformation). *Denote the translation dynamics: $\hat{\pi}^{tra} = \begin{bmatrix} I_m & \boldsymbol{v} \\ \mathbf{0} & 1 \end{bmatrix}$ and its Lie group $H \subset \mathrm{E}(m)$, where $\mathrm{E}(m)$ is an Euclidean group. They satisfy the conditions in Proposition 1. Denote the scaling dynamics: $\hat{\pi}^{sca} = \begin{bmatrix} \mathrm{diag}(\boldsymbol{\gamma}) & \mathbf{0} \\ \mathbf{0} & 1 \end{bmatrix}$ and its Lie group $H \subset \mathrm{Sim}(m)$, where $\mathrm{Sim}(m)$ is a similarity group. They satisfy the conditions in Proposition 1.*

*Proof.* In the case of translation transformation $\hat{\pi}^{tra}$, for identity matrices, we have $I_m I_m = I_m I_m$. Also, we have $I_m \times \boldsymbol{v} = \boldsymbol{v}$ for $\forall \boldsymbol{v} \in \mathbb{R}^m$. Therefore translation transformation $\hat{\pi}^{tra}$ satisfies the condition in Proposition 1. In the case of scaling transformation $\hat{\pi}^{sca}$, for diagonal matrices, we have $\mathrm{diag}(\boldsymbol{\gamma}_1)\mathrm{diag}(\boldsymbol{\gamma}_2) = \mathrm{diag}(\boldsymbol{\gamma}_2)\mathrm{diag}(\boldsymbol{\gamma}_1)$. Also, we have $\mathrm{diag}(\boldsymbol{\gamma}_1) \times \mathbf{0} = \mathbf{0}$. Therefore scaling transformation $\hat{\pi}^{sca}$ satisfies the condition in Proposition 1. $\qquad\square$

## A.5   Proof of Corollary 3

**Corollary 4** (Equivariance of Second-Order Composed Transformations). *The second-order composition of the planar rotation, translation, and scaling transformations defined in Corollaries 1–2 satisfies the commutativity conditions in Proposition 1. Specifically, for any $\hat{\pi}^{(1)}, \hat{\pi}^{(2)} \in \{\hat{\pi}^{rot}, \hat{\pi}^{tra}, \hat{\pi}^{sca}\}$, we have that $\hat{\pi}^{(1)}\hat{\pi}^{(2)}$ and its Lie group $H \subset \mathrm{Aff}(m)$ satisfy the conditions in Proposition 1.*

*Proof.* To show that $\hat{\pi}^{(1)}\hat{\pi}^{(2)}$ satisfies the commutativity conditions in Proposition 1, we need to show that

$$\hat{\pi}^{(1)}(g_1)\hat{\pi}^{(2)}(g_2)\hat{\pi}^{(1)}(h_1)\hat{\pi}^{(2)}(h_2) = \hat{\pi}^{(1)}(h_1)\hat{\pi}^{(2)}(h_2)\hat{\pi}^{(1)}(g_1)\hat{\pi}^{(2)}(g_2), \forall g_1, g_2, h_1, h_2 \in H.$$

Since $\hat{\pi}^{(1)}, \hat{\pi}^{(2)} \in \{\hat{\pi}^{rot}, \hat{\pi}^{tra}, \hat{\pi}^{sca}\}$ which already satisfy the commutativity conditions as shown in Corollaries 1–2, we have

$$\hat{\pi}^{(1)}(g_1)\hat{\pi}^{(1)}(h_1) = \hat{\pi}^{(1)}(h_1)\hat{\pi}^{(1)}(g_1), \quad \forall g_1, h_1 \in H$$

$$\hat{\pi}^{(2)}(g_2)\hat{\pi}^{(2)}(h_2) = \hat{\pi}^{(2)}(h_2)\hat{\pi}^{(2)}(g_2), \quad \forall g_2, h_2 \in H$$

Thus, the initial condition is reduced to showing that $\hat{\pi}^{(1)}$ and $\hat{\pi}^{(2)}$ can be commutative, i.e.,

$$\hat{\pi}^{(1)}(g)\hat{\pi}^{(2)}(h) = \hat{\pi}^{(2)}(h)\hat{\pi}^{(1)}(g), \quad \forall g, h \in H$$

Consider two transformations $\hat{\pi}^{(1)}, \hat{\pi}^{(2)} \in \{\hat{\pi}^{rot}, \hat{\pi}^{tra}, \hat{\pi}^{sca}\}$, their general affine transformation form is:

$$\hat{\pi}^{(k)} = \begin{bmatrix} A^{(k)} & \boldsymbol{b}^{(k)} \\ \boldsymbol{0} & 1 \end{bmatrix}, \quad \text{for } k = 1, 2,$$

where $A^{(k)} \in \mathbb{R}^{m \times m}$ and $\boldsymbol{b}^{(k)} \in \mathbb{R}^m$. We have:

$$\hat{\pi}^{(1)}\hat{\pi}^{(2)} = \begin{bmatrix} A^{(1)}A^{(2)} & A^{(1)}\boldsymbol{b}^{(2)} + \boldsymbol{b}^{(1)} \\ \boldsymbol{0} & 1 \end{bmatrix}, \quad \text{and} \quad \hat{\pi}^{(2)}\hat{\pi}^{(1)} = \begin{bmatrix} A^{(2)}A^{(1)} & A^{(2)}\boldsymbol{b}^{(1)} + \boldsymbol{b}^{(2)} \\ \boldsymbol{0} & 1 \end{bmatrix}.$$

According to Proposition 1, commutativity holds if:

$$A^{(1)}A^{(2)} = A^{(2)}A^{(1)} \quad \text{and} \quad A^{(1)}\boldsymbol{b}^{(2)} + \boldsymbol{b}^{(1)} = A^{(2)}\boldsymbol{b}^{(1)} + \boldsymbol{b}^{(2)}.$$

We now verify these two conditions for all combinations:

  (i) **Linear Part:** $A^{(1)}A^{(2)} = A^{(2)}A^{(1)}$

- If both are diagonal scaling matrices, this holds since diagonal matrices commute.
- If both are planar rotations in the same subspace, they commute by the abelian property of SO(2).
- If either matrix is $I_m$ (as in translation), the product trivially commutes.
- If one is scaling (diagonal) and the other is a planar rotation, they commute within the rotation subspace or when scaling is isotropic.

  (ii) **Translation Part:** $A^{(1)}\boldsymbol{b}^{(2)} + \boldsymbol{b}^{(1)} = A^{(2)}\boldsymbol{b}^{(1)} + \boldsymbol{b}^{(2)}$

- If both transformations are translations, $A^{(1)} = A^{(2)} = I_m$, and the equality holds trivially.
- If one is translation and the other is rotation or scaling, we have:

$$A^{(1)}\boldsymbol{b}^{(2)} = \boldsymbol{b}^{(2)}, \quad A^{(2)}\boldsymbol{b}^{(1)} = \boldsymbol{b}^{(1)},$$

  since translation has $A^{(k)} = I_m$, satisfying the equality.
- If both are non-translation (rotation or scaling), $\boldsymbol{b}^{(1)} = \boldsymbol{b}^{(2)} = \boldsymbol{0}$, and the equality holds.

In all cases, the matrix product and translation terms satisfy the required commutation conditions. Therefore,

$$\hat{\pi}^{(1)}\hat{\pi}^{(2)} = \hat{\pi}^{(2)}\hat{\pi}^{(1)},$$

and the composed transformations satisfy the conditions of Proposition 1.  □

# B Tasks, Data Generation, and Preprocessing

## B.1 Interpolation and Extrapolation Tasks

We adopt a standard formulation consistent with prior work on continuous-time and irregularly sampled time series, such as Latent ODE, Contiformer, RNN-$\Delta_t$, etc. The setup is as follows:

(1) Interpolation. Given a time series with time points $(t_0, \cdots, t_N)$, we condition on the subset of points from $(t_0, \cdots, t_N)$ with a data drop rate (30%, 60%, 90% in Experiments) and reconstruct the full set of points in the same time interval.

(2) Extrapolation. We split the time series into two parts $(t_0, \cdots, t_{N/2})$ and $(t_{N/2}, \cdots, t_N)$. We input the first half of the time series and predict the second half. We apply the same random drop procedure only on the input half, and the model is tasked with predicting the entire future segment beyond the observed range.

## B.2 Complex ODE systems

**Spiral dataset**. We generate 80 trajectories with 60 timesteps. The spiral system is a two-dimensional system with rotation and scaling symmetries, characterized by the infinitesimal generator $v = x_2 \partial_1 - x_1 \partial_2$ (i.e., $\begin{bmatrix} 0 & 1 \\ -1 & 0 \end{bmatrix}$ in matrix terms). It is governed by:

$$\begin{cases} \dot{x}_1 = -0.1x_1 - x_2, \\ \dot{x}_2 = x_1 - 0.1x_2, \end{cases} \tag{17}$$

We also create more complex ODE systems with nonlinear symmetries.

**Glycolytic Oscillator**. We generate 100 trajectories, each with 200 timesteps. The two-dimensional Glycolytic Oscillator system [73] models a biochemical process governed by a pair of coupled ODEs with complex cubic nonlinear interactions. We adopt the same parameter settings as used in [37]. The governing equations are given by:

$$\begin{cases} \dot{x}_1 = 0.75 - 0.1x_1 - x_1 x_2^2, \\ \dot{x}_2 = 0.1x_1 - x_2 + x_1 x_2^2, \end{cases} \tag{18}$$

**Lotka-Volterra System**. We generate 100 trajectories, each with 200 timesteps. The two-dimensional Lotka-Volterra System is a classical model in population dynamics that describes the nonlinear interactions between predator and prey species. It is governed by the following coupled ODEs:

$$\begin{cases} \dot{x}_1 = 0.1x - 0.02x_1 x_2, \\ \dot{x}_2 = 0.01x_1 x_2 - 0.3x_2, \end{cases} \tag{19}$$

## B.3 Power system datasets

**Electricity consumption**. We gather load consumption of Flores, Azores Islands in the year of 2008 [51]. The profile contains 366 days' load data with a sampling interval of 10min. We treat each day's data (144 points) as a trajectory and gather 100 trajectories.

**Solar energy generation**. We use a publicly available photovoltaic (PV) dataset [54] that records sequential solar power generation measurements. The data was collected in 2017 from a weather station located on the Gaithersburg, Maryland campus of the National Institute of Standards and Technology (NIST), with a sampling interval of 1 minute. We select data from four different locations and treat each day at each location as an individual trajectory, resulting in a total of 124 trajectories for a one-month period. To focus on periods with active solar generation, we remove nighttime measurements with zero output and extract 100 valid time steps per trajectory.

**Event measurements**. Following [74, 75], we simulate power system events using the commercial-grade Positive Sequence Load Flow (PSLF) software [76] developed by General Electric (GE). The simulations are based on the publicly available Illinois 200-node system [77]. Each simulation spans 4 seconds with a sampling interval of 33.33 milliseconds, capturing high-resolution system dynamics.

We extract measurements from 100 selected nodes, treating each node's time series as a separate trajectory. This results in 100 trajectories, each containing 100 time steps. Each trajectory contains 10-dimensional data: voltage magnitude, voltage angle, current magnitude, current angle, frequency, rate of change of frequency, rotor speed, rotor angle, active power, and reactive power.

### B.4  Weather system dataset

**Air quality.** We consider air quality data from the UCI repository [78], which contains the hourly responses of a gas multisensor device deployed on the field in an Italian city. Specifically, we use the `C6H6(GT)` feature, which records the hourly averaged benzene concentration (in $\mu$g/m$^3$). We select the period from October 1, 2004 to December 13, 2004, containing 73 days of measurements without outliers. We treat each day's data (24 points) as a trajectory and gather 92 trajectories.

### B.5  Biomedical system dataset

The signals correspond to electrocardiogram (ECG) shapes of heartbeats for the normal case and the cases affected by different arrhythmias and myocardial infarctions. We employ the ECG200 in [59]. The training data contains 100 trajectories, each with 97 timesteps.

## C  Introduction of Baseline Methods in Experiments

The following baseline methods are used. For interpolation, we have: (1) **EqSINDy** [37]. EqSINDy adds equivariance regularization to SINDy method to discover the symbolic equations for the ODE data. The equivariance relation is discovered through a GAN-based framework [40]. (2) **EqGP** [37]. Similar to EqSINDy, the equivariance relation is added to a Generic programming to discover symbolic ODE equations. (3) **RNN-$\Delta_t$** [79]. The time difference between every two observations, i.e., $\Delta_t$, is introduced to a classic RNN model. (4) **ODE-RNN** [18]. ODE-RNN can be directly utilized for dynamic learning in an encoder-only framework. (5) **Neural CDE (NCDE)** [17]. Neural CDE creates a continuous data path to control the evolution of the state's ODE flow. (6) **ContiFormer** [80]. ContiFormer is a Transformer-based with a continuous attention mechanism to extract feature flows at arbitrary times. (7) **Latent ODE** [18]. The model is illustrated in Section 2.3. For extrapolation, in addition to the above methods, we introduce the state-of-the-art Transformer-based methods for time-series forecasting: (8) **Informer** [81]. Informer is a transformer variant tailored for time-series forecasting, using a probabilistic sparse self-attention mechanism to reduce computation and improve efficiency. (9) **Autoformer** [82]. Autoformer enhances time-series prediction by introducing series decomposition and autocorrelation-based attention, enabling it to better model trend and seasonal patterns. As Informer and Autoformer can't process irregularly-sampled data, we conduct spline interpolation to pre-process input data for extrapolation tasks. For fair comparisons, we maintain roughly the same size of the hidden state, number of layers, and units among different methods. Architecture details and training time are reported in Appendix D and Appendix E.3, respectively.

## D  Model Architecture and Hyper-parameters

**Computing infrastructure**. All experiments were run in Python 3.12 on a Mac machine equipped with an Intel Core i5 processor (3.1 GHz) and 8 GB of RAM.

For fair comparisons, we maintain roughly the same size of the hidden state, number of layers, and units. In particular, we make the dimension of the latent space $\mathcal{Z}$, i.e., $m$, to be the same for Latent ODE and latent MoS. We also restrict them to having the same ODE-RNN encoder. The specific architectures and hyper-parameters are described as follows.

**Encoder**. Similar to Latent ODE [18], we employ an ODE-RNN as the encoder, as illustrated in Equation (4). In this ODE-RNN, we set the GRU module the same as the commonly used GRU baseline model, where the update gate, reset gate, and state transition module have two hidden layers with the hidden unit number to be $m$ (specific values are shown in the following parts). The ODE function module also has two hidden layers with the hidden unit number to be $m$. The ODE solver used to solve the ODE-RNN is the fourth-order Runge–Kutta method ("rk4").

**Decoder**. The decoder is composed of the parameterized Lie group representation and the gating neural network. For each of the network, we still consider two hidden layers with the number of neuron to be $m$. Their output dimensions are directly related to $m$ and $K$, which have been illustrated in the main section.

**Multi-level MoS**. In our decoder, we consider $S = 2$ levels for multi-level MoS (see Section 3.3). Moreover, we set $L^1 = 2$ and $L^2 = 5$ to extract multi timescale equivariances.

**Output layers**. There are two hidden layers with the number of units to be $m$.

**Activation function and learning rate.** We utilize Tanh as the activation function, similar to Latent ODEs. We set the learning rate to be $0.001$.

**Restrictions on each of the Lie group actions.** To enable smooth transitions, we enforce certain restrictions on the Lie group transition during $\Delta t$. Specifically, for rotational symmetry, we assume $\theta$ in $R_\theta$ to be within $[-0.6, 0.6]$. For translation symmetry, we assume the maximum norm of $v(\boldsymbol{z}(t), t)$ to be less than $0.001$. For scaling symmetry, we assume $-1.5 \leq \gamma(\boldsymbol{z}(t), t) \leq 1.5$. These restrictions can be easily achieved by using Tanh activation. Further, to avoid scaling and translation dynamics leading to vanishing or exploding latent vectors, we rescale the norm of $\boldsymbol{z}(t_i)$ into $(0.5, 1.5)$.

**Gating mechanism**. We encourage the exploration in the gating mechanism by assuming a warmup period where we don't restrict any sparsity to the gating values. After $10$ epoch for the warmup, we select the top $K_0 \leq K$ experts to build the mixture of latent flows.

**Hyper-parameters** We establish two important hyper-parameters for different datasets: the dimension of the latent space $m$ and the number of experts $K_0$ for non-zero outputs. Specifically, they are shown in the following table.

Table 2: Hyper-parameter settings for different datasets.

| System | Spiral | Glycolytic | Lotka | Load | Solar | Power event | Air quality | ECG |
|---|---|---|---|---|---|---|---|---|
| Latent dimension $m$ | 15 | 15 | 15 | 15 | 15 | 30 | 15 | 15 |
| Non-zero gate numbers $K_0$ | 2 | 2 | 4 | 4 | 4 | 6 | 4 | 6 |

Table 3: Test Mean Squared Error (MSE) ($\times 10^{-2}$) for interpolation tasks under varying drop rates.

| Data | Drop | EqSINDy | EqGP | RNN-$\Delta_t$ | ODE-RNN | NCDE | Contiformer | Latent ODE | **Latent MoS** |
|------|------|---------|------|----------------|---------|------|-------------|------------|----------------|
| Spiral | 90% | 11.21 | 13.88 | 9.39 | 11.47 | 16.30 | 3.23 | 2.30 | **1.93** |
| | 60% | 5.67 | 6.97 | 4.02 | 11.01 | 11.39 | 1.66 | 1.23 | **0.97** |
| | 30% | 2.12 | 2.13 | 2.05 | 10.97 | 8.03 | 1.32 | 1.14 | **0.74** |
| Glycolytic | 90% | 14.45 | 23.09 | 14.28 | 14.08 | 31.90 | 4.66 | 11.24 | **0.33** |
| | 60% | 6.38 | 10.58 | 7.76 | 13.54 | 22.82 | 2.89 | 2.68 | **0.21** |
| | 30% | 2.74 | 4.26 | 2.34 | 13.27 | 17.44 | 0.43 | 1.64 | **0.18** |
| Lotka | 90% | 13.45 | 10.98 | 11.22 | 3.75 | 38.13 | 2.21 | 2.63 | **0.53** |
| | 60% | 7.88 | 4.77 | 6.26 | 3.34 | 28.21 | 0.92 | **0.22** | 0.30 |
| | 30% | 2.42 | 1.61 | 2.43 | 3.26 | 20.55 | 0.91 | 0.09 | **0.06** |
| Load | 90% | 18.98 | 16.79 | 8.53 | 3.51 | 14.88 | 6.61 | 2.76 | **2.30** |
| | 60% | 8.45 | 7.74 | 3.13 | 2.37 | 18.25 | 4.50 | **0.97** | 1.10 |
| | 30% | 5.21 | 4.37 | 1.37 | 2.23 | 20.78 | 4.45 | **0.89** | 0.90 |
| Solar | 90% | 15.89 | 13.79 | 40.21 | 5.69 | 32.07 | 11.96 | 4.01 | **3.10** |
| | 60% | 13.57 | 14.23 | 16.09 | 3.94 | 24.39 | 12.06 | 2.65 | **1.23** |
| | 30% | 10.11 | 10.87 | 6.43 | 3.78 | 14.37 | 12.21 | 1.73 | **0.98** |
| Power event | 90% | 8.44 | 6.89 | 12.95 | 5.71 | 15.21 | 5.09 | 5.41 | **4.06** |
| | 60% | 6.58 | 4.53 | 8.82 | 5.22 | 14.52 | 4.34 | 4.17 | **3.65** |
| | 30% | 3.28 | 3.75 | 4.34 | 5.13 | 12.02 | 3.40 | 4.01 | **3.14** |
| Air quality | 90% | 15.89 | 18.70 | 17.38 | 4.76 | 26.70 | 7.53 | 4.36 | **3.30** |
| | 60% | 15.23 | 15.99 | 12.18 | 3.27 | 14.61 | 8.12 | 3.16 | **2.24** |
| | 30% | 10.98 | 14.03 | 6.23 | 2.88 | 8.78 | 6.73 | 2.43 | **1.53** |
| ECG | 90% | 13.41 | 10.09 | 10.42 | 6.37 | 13.56 | 5.02 | 2.87 | **1.15** |
| | 60% | 5.62 | 3.21 | 4.99 | 5.52 | 11.03 | 3.17 | 1.42 | **0.56** |
| | 30% | 3.08 | 1.05 | 2.28 | 5.41 | 9.92 | 3.07 | 1.35 | **0.34** |

# E  Supplementary Experimental Results

## E.1  Table of Interpolation Results

We present the tabular results for Fig. 3 as follows. Contiformer, Latent ODE, and Latent MoS can learn the continuous latent ODE in the decoding process, thus achieving the best performances. Among them, Latent MoS leverages geometric priors to structure the latent trajectories. Across all datasets, Latent MoS achieves relative improvements ranging from $15\%$ to $97\%$.

### E.2 Predicted Curve Visualization

We visualize the predicted trajectories of three complex ODE systems, including the Spiral system, the Glycolytic Oscillator, and the Lotka-Volterra system, under low-resolution data conditions with a drop rate of 90%. The ground truth trajectories are shown as black curves, the observed low-resolution data points are plotted as blue dots, and the predicted trajectories from both the Latent ODE model and our proposed Latent MoS model are illustrated as orange dashed curves. As shown in Fig. 6, the Latent MoS model consistently provides more accurate reconstructions of the underlying dynamics, which we attribute to its enhanced capability to preserve hidden symmetries in the data.

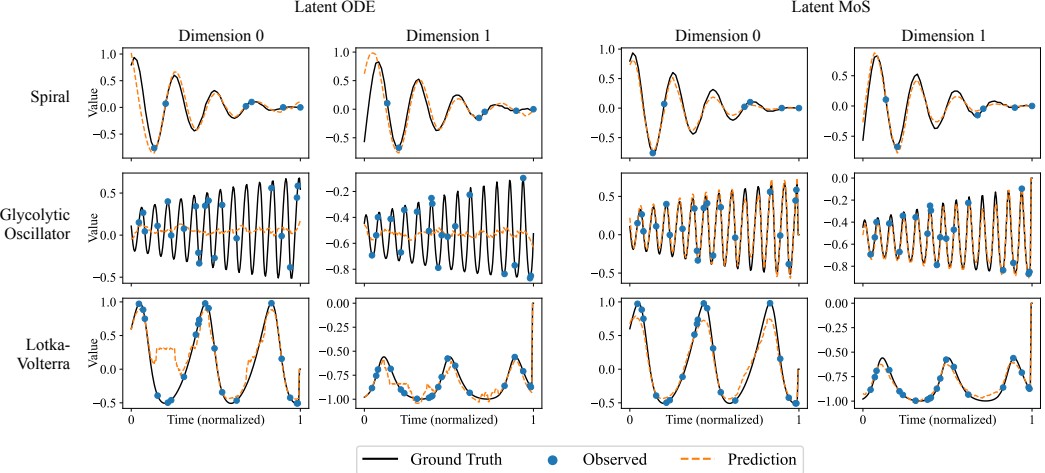

Figure 6: Comparison of ground truth trajectory and predicted trajectory for three complex ODE systems (Spiral, Glycolytic Oscillator, and Lotka-Volterra) under low-resolution data with a 90% drop rate.

### E.3 Training Time

We report the training time for different models in the following table. The results show that our methods can achieve relatively moderate training time, comparable to Latent ODE and ODE-RNN. We eliminate the results EqSINDy and EqGP methods because they require separate symmetry discovery and symbolic regression processes.

Table 4: Training time (minutes) for different systems and models.

| Data | RNN-$\Delta_t$ | ODE-RNN | NCDE | Contiformer | Informer | Autoformer | Latent ODE | **Latent MoS** |
|------|------|---------|------|-------------|----------|------------|------------|----------------|
| Spiral | 4.9 | 6.5 | 46.4 | 111.2 | 70.7 | 58.5 | 7.3 | 8.4 |
| Glycolytic | 7.7 | 10.4 | 81.2 | 162.5 | 121.9 | 91.7 | 12.2 | 15.5 |
| Lotka | 9.4 | 10.2 | 74.5 | 193.3 | 125.9 | 90.3 | 12.6 | 15.4 |
| Load | 7.5 | 10.1 | 80.4 | 168.8 | 120.6 | 98.5 | 12.0 | 13.4 |
| Solar | 11.4 | 13.4 | 83.6 | 204.1 | 150.6 | 112.5 | 15.7 | 18.1 |
| Power event | 15 | 19.4 | 118.8 | 317.4 | 211.6 | 165.6 | 21.2 | 25.4 |
| Air quality | 6.0 | 6.7 | 53.0 | 116.9 | 85.8 | 63.2 | 8.4 | 10.0 |
| ECG | 6.8 | 9.2 | 59.7 | 153.3 | 103.9 | 76.4 | 10.4 | 13.3 |

