# OpenReview forum: "Latent Mixture of Symmetries for Sample-Efficient Dynamic Learning"
_NeurIPS.cc/2025/Conference — NeurIPS 2025 poster_

### Official Review · Reviewer_UuCF · 2025-07-02

**Clarity:** 4
**Significance:** 2
**Originality:** 3
**Rating:** 4
**Confidence:** 3

**Summary:**

This paper proposes combining MoEs with Neural ODEs to learn latent dynamics which respect symmetry. The objective is similar to symmetry discovery, but avoids common limitations such as error propagation and limited expressiveness of multiple symmetry groups. The authors use a mixture of latent flows where each flow is governed by a different Lie group and use a encoder-decoder architecture for nonlinear group actions on the data. Experiments on many different domains with ODE dynamics and real-world data show that their method outperforms baselines.

**Questions:**

- Are the symmetry groups known a priori? Or can the method learn (perhaps through the gating mechanism) subgroups of the general linear/affine group.
- How are learned symmetries preserved across time steps? Would this method correct learn the Lie point symmetries of the governing equations?

**Ethical Concerns:**

["NO or VERY MINOR ethics concerns only"]

**Final Justification:**

The author's response has changed my view of the paper, in particular the motivation behind using a mixture of symmetry groups (which produces a nonvalid symmetry) for specific dynamics systems and real-world datasets.

**Limitations:**

See weaknesses and questions.

**Quality:**

2

**Strengths And Weaknesses:**

## Strengths
- The authors provide various theoretical analyses on how Latent MoS can learn several different symmetry groups.
- The authors experiment on several datasets, including real-world domains such as residential electricity consumption, solar energy, etc.

## Weaknesses
- I feel like the core flaw of the proposed approach is the linear mixing/weighted average (Equation 8). This seems to go against how symmetries should be composed together (composition/matrix multiplication). It's not clear to me how this method really does learn both symmetries with this mixing technique and it seems to in fact break the group structure, since the mixture isn't a valid group element. Can the authors clarify?
- Furthermore, the ODE-RNN structure doesn't seem to use the same mixture or symmetry groups for the latent dynamics at all timesteps. Why would a symmetry be present at a certain timestep but not present at all other timesteps? The proposed method doesn't seem to preserve the mixing weights from previous timesteps.
- LaLiGAN (Reference [38] Yang et al. 2023) and perhaps a downstream dynamics model should be included as baseline. It seems to have the same objective (learning latent symmetries) as this paper.

---

> ### Author Rebuttal · Authors · 2025-07-30
>
> **Q1. Flaws due to the mixture model**.
>
> It is correct that the mixture in Equation (8) does not define a valid Lie group action. This is our deliberate and innovative design to **model the system dynamics**.
>
> Our design only requires that the **components of the latent vector** have symmetry transformations. This is because in Introduction and Section 2.1, we discussed that many real-world systems are influenced by multiple latent factors acting simultaneously, each evolving under a distinct Lie group action. For example, damping (scaling) and oscillation (rotation) forces co-exist in many mechanical and electrical systems. These effects operate concurrently and cannot be captured by a single group action or composition. Instead, they give rise to a superposition of latent flows, each governed by its own symmetry structure. Our model captures this behavior by treating each expert as a valid Lie group action and using a gating network to assign weights that combine their effects. While the resulting update (Equation 8) is not itself a group action, it provides a principled and expressive way to model complex system dynamics. **In general, Equation (8) defines the system's latent dynamics, but doesn't need to be a valid Lie transformation**.
>
> For example, in our experiments with power event datasets, the results show that the gating weights for rotation and scaling are 0.83 and 0.17, respectively. This indicates that both **intrinsic oscillations and the extrinsic damping force** from devices like a power grid stabilizer (PSS) are simultaneously captured. Finally, thanks to our highly expressive yet symmetry-preserving design, Latent MoS performs strongly on complex real-world dynamical systems such as power events, ECG, and air quality. It achieves 30–60\% reductions in MSE compared to state-of-the-art methods like Latent ODE.
>
> When the gating network concentrates on a single expert, **our model reduces to a special case** where the dynamics are governed by a single (nonlinear) Lie group action.  For example, our answers in Q5 show how the synthetic ODE datasets lead to single rotation × scaling symmetry.
>
> **Q2. Time range of the symmetry.**
>
> This question directly echoes another key contribution we discuss in Section 2.1: short- and long-term equivariance in dynamical systems. Unlike classical equivariant models that enforce fixed symmetry globally, our method allows the effective symmetry structure to vary and persist over flexible time horizons.
>
> In our model, **the mixing weights and symmetry transformations are held fixed within predefined temporal intervals**, which ensures consistent symmetry behavior over that range. Specifically, as shown in Equation (9) in Section 3.2, for each interval indexed by $l$, we define a start time $t^{(l)}$ and use the tuple $(t^{(l)},\boldsymbol{z}(t^{(l)}))$
> as input to neural networks that generate both the gating weights and the symmetry matrices. This construction ensures that within a given interval, **the same mixture of symmetry-preserving flows** is applied. Moreover, as described in Algorithm 1 (Section 3.3), our multi-level Latent MoS architecture introduces variable interval lengths across different temporal resolutions. This enables the model to **flexibly capture short-term or long-term equivariance**, depending on the system's structure and data complexity.
>
> For example, in our load forecasting dataset with 144 time points per day (i.e., 10-minute resolution), we define two temporal levels: a short-term level spanning 3 hours (18 points) and a long-term level covering the full day. The results show that the 3-hour interval exhibits a dominant scaling symmetry (gating weight = 0.96), reflecting local growth or decay. In contrast, the full-day interval reveals a dominant rotation symmetry (gating weight = 0.91), indicating a cyclic structure in the dynamics, such as daily progression through peak and off-peak periods.
>
> Finally, in our ODE-based synthetic datasets, even when we define both short- and long-term levels, the gating weights and learned symmetries remain nearly the same values across levels, demonstrating that the system exhibits a global symmetry structure. Specific results are illustrated in our response to Q5.
>
> **Remark: symmetry breaking**. As discussed in our responses to Q1 and Q2, one of the key contributions of Latent MoS is its ability to model not only systems with global symmetries but also those that exhibit symmetry breaking, i.e., the assumption of strict, global invariance does not hold uniformly over time. Our work identifies symmetry breaking in two orthogonal axes: **a mixture of groups and a short- or long-time dependence** (see Section 2.1). This breaking changes the restrictive assumption of a fixed affine group. The mixture of groups appears in literature [Kim et al., 2023], and the locally adaptable symmetry breaking is discussed in [Wang et al., 2022; Wang et al., 2023; Ouderaa et al., 2022]. They are independently discussed, but our method addresses both in a unified framework.
>
> To model symmetry breaking, past methods (1) utilize learnable or non-stationary weights [Wang et al., 2022], (2) add soft regularization [Kim et al., 2023; Finzi et al., 2021], or (3) construct a symmetry-breaking set as auxiliary inputs [Xie & Smidt, 2024]. Our model adopts method (1) without the need to construct additional terms or sets.
>
> **Q3. Baseline for symmetry discovery and dynamic learning**.
>
> The reason we didn't include this baseline is that this type of method has strong requirements for data quality, but we mainly evaluate realistic sample-inefficient cases. For example, GAN and SINDy in  [Yang et al., 2024] both require significant samples. Naturally, these baselines perform worse than other ODE-structured models like Latent ODE. As the reviewer asked, we conducted the experiments using the official code from [Yang et al., 2024] and tested two recent symmetry-discovery methods: EquivSINDy-r (SINDy-based) and EquivGP-r (Genetic Programming-based). They all require LaLiGAN to discover symmetry, which is further used in SINDy or GP as regularization. We report interpolation errors under a 30\% data drop rate. For the Spiral, Glycolytic, and Lotka systems, the MSEs ($\times 10^{-2}$) are as follows: EquivSINDy-r: 2.12, 2.74, 2.42; EquivGP-r: 2.13, 4.26, 1.61, all of which are around 4× to 10× larger than the results achieved by our Latent MoS. Similar trends hold across other datasets.  These findings highlight that existing symmetry-discovery methods based on GAN priors and symbolic regressors (SINDy/GP) are highly sensitive to data quality and quantity. In contrast, our method remains robust under sparse and limited trajectories, which are typical in dynamical systems.
>
> **Q4. Symmetry group as a priori**.
>
> Unlike many past methods [Weiler \& Cesa, 2019;Cohen et al., 2019a] that require groups as a prior, latent MoS learns symmetry from data via the gating mechanism. Each expert in our model corresponds to a subgroup of the affine group (e.g., rotation, translation, etc.), and the gating network assigns weights to form a data-driven mixture of these symmetry-preserving flows. This design allows the model to adaptively select and combine latent symmetry structures at different times and regions, offering high expressive power for modeling complex and potentially symmetry-breaking systems.
>
> **Q5. Learn Lie point symmetry**.
>
> Our method can learn the Lie point symmetry. In our experiment, the physical systems in Fig. 2 contain global Lie point symmetries. The visualization of the latent trajectory indicates that one of our latent components accurately captures the dominant symmetry. For example, the Glycolytic system is governed by rotation × scaling symmetry.  In our result, the gating weights are highly concentrated: the dominant expert, i.e., a second-order transformation (rotation × scaling), receives a weight of 0.97, while the next highest (scaling × rotation) receives 0.025, and the remaining experts collectively contribute less than 0.005. Thanks to this accurate selection, the model achieves an extremely low MSE of 0.0033. The low interpolation/extrapolation error indicates the true symmetry is approximately captured.
>
> We feel that it's good to test if the model can directly estimate the ground-truth symmetry transformation. Hence, we **conducted an additional experiment**. We set the latent dimension $m=2$ and fixed both the encoder and decoder to identity mappings. This eliminates the possibility of the encoder and decoder jointly transforming the latent dynamics into an equivalent—but potentially different from the true governing equation.
>
> Then, for the spiral dataset, Latent MoS recovers a scaling factor of 0.98 and a rotation matrix R = [0.968, -0.248; 0.248, 0.968], which closely matches the true dynamics used in data generation. **The average error compared to the ground truth symmetry is less than 0.05, and the gating weight for this expert (rotation × scaling) is 1**. This experiment demonstrates that Latent MoS can faithfully recover interpretable symmetry structures (or their equivalent forms) directly from data.
>
> Finally, we note that restricting the model capacity to learn exact the true symmetry transformation is uncessary because (1) real-world systems can have changing symmetries and (2) the true transformation matrix can be represented by many equivalent forms in our encoder-decoder structure. The test MSE in interpolation/extrapolation is a more appropriate mearue for our dynamic learning problem.

---

### Official Review · Reviewer_KeLt · 2025-07-02

**Clarity:** 3
**Significance:** 3
**Originality:** 3
**Rating:** 4
**Confidence:** 2

**Summary:**

This paper addresses the problem of dynamic learning under data scarcity. The authors introduce the Latent Mixture of Symmetries (MoS) framework based on the Mixture-of-Experts (MoE) approach, efficiently preserving local equivariance through structured transformations. Utilizing a gating mechanism derived from MoE, MoS adaptively selects the most appropriate local Lie group symmetries. Extensive evaluations on a range of synthetic and real-world dynamical systems demonstrate promising improvements compared to several baseline methods.

**Questions:**

1. How sensitive is Latent MoS to the accurate identification of underlying symmetry groups? As indicated by ablation studies, including unnecessary symmetries can negatively impact performance. Could the authors comment on any conceptual connections between this scenario and overcomplete dictionary learning, considering symmetries analogous to dictionary items?

2. Could the authors clarify how the interpolation and extrapolation tasks are precisely defined?

3. In Figure 3 (Lotka System - Drop rate 30%), the blue bar appears to be missing. Could the authors provide clarification or correct this oversight?

**Ethical Concerns:**

["NO or VERY MINOR ethics concerns only"]

**Final Justification:**

I am happy with the authors' rebuttal and would keep my rating of borderline accept.

**Limitations:**

yes

**Quality:**

3

**Strengths And Weaknesses:**

### Strengths
1. The concept of modeling different local symmetries within a single trajectory is appealing, clearly motivated, and innovative.
2. The paper is well-structured, presenting detailed theoretical foundations supported by extensive empirical evaluations.
3. The experimental results consistently indicate superior performance of the proposed approach compared to existing baselines.

### Weaknesses
See Questions below.

---

> ### Author Rebuttal · Authors · 2025-07-30
>
> **Q1. Sensitivity and relations to dictionary learning**.
>
> We provide an additional sensitivity analysis to assess the impact of including irrelevant symmetry groups. Specifically, we consider the Glycolytic system with a 90\% drop rate and vary the number of available symmetry experts from 1 to 9, while always including the necessary one (rotation × scaling). The resulting MSEs ($\times 10^{-2}$) for 1 to 9 experts are: 0.19, 0.20, 0.20, 0.25, 0.27, 0.28, 0.31, 0.33, and 0.33, respectively. These results show that introducing irrelevant experts can slightly degrade performance; however, even with all experts, the model still achieves a low MSE of 0.0033. In this setting, the learned gating weights remain concentrated: the dominant expert (rotation × scaling) receives a gating weight of 0.97, while others are negligible (e.g., 0.025 for scaling × rotation and $<0.005$ for the remaining). This demonstrates that the gating mechanism effectively identifies the correct symmetry. Nonetheless, we appreciate the reviewer’s suggestion regarding overcomplete dictionary learning and recognize that advanced techniques from that literature could further improve MoE selection in the presence of redundant experts.
>
>
> Overcomplete dictionary learning is a long-standing topic that seeks to represent data as sparse linear combinations of basis atoms, enabling compact and interpretable representations. Recent theoretical advances have introduced provable algorithms for learning incoherent dictionaries [Arora et al., 2014; Sulam et al., 2022] and regularization strategies for identifiability and convergence guarantees [Sun et al., 2025]. Moreover, recent works [Wang et al., 2022; Mudide et al., 2025] have uncovered deep connections between dictionary learning and Mixture-of-Experts (MoE) models, including our Mixture-of-Symmetries (MoS): each expert can be viewed as a dictionary atom, and the gating network plays an analogous role to sparse coding, selecting a minimal set of active atoms. This conceptual link suggests that advances in dictionary learning, such as novel regularization and pruning, could inspire improved training and regularization techniques for MoS. We leave this direction as a promising avenue for future work.
>
> **Q2. Interpolation and Extrapolation clarification**.
>
> We adopt a standard formulation consistent with prior work on continuous-time and irregularly sampled time series, such as Latent ODE, Contiformer, RNN-$\Delta\_t$, etc. For example, Appendix 1 in Laten ODE [Rubanova et al. 2019] gives a good description. The setup is as follows:
>
> (1) Interpolation. Given a time series with time points $(t\_0,\cdots,t\_N)$, we condition on the subset of points from $(t\_0,\cdots,t\_N)$ with a data drop rate (30\%, 60\%, 90\% in Experiments) and reconstruct the full set of points in the same time interval.
>
> (2) Extrapolation. We split the time series into two parts $(t_0,\cdots,t_{N/2})$ and $(t_{N/2},\cdots,t_N)$. We input the first half of the time series and predict the second half. We apply the same random drop procedure only on the input half, and the model is tasked with predicting the entire future segment beyond the observed range.
>
> Additional details regarding dataset sizes, number of trajectories, and time resolution for each system are provided in Appendix B in our paper.
>
> **Q3. Figure issue**.
>
> We thank the reviewer for pointing out the apparent omission of the blue bar in Figure 3 (Lotka system – Drop rate 30%). This issue was caused by an unintentional use of the plotting command "ax.set_ylim(0.001, 0.4)" during figure generation for better visual look, which truncated values below 0.001. As a result, both Latent ODE (0.0009) and our proposed Latent MoS (0.0006), which achieve significantly lower MSEs compared to other methods (e.g., NCDE: 0.2055), were visually suppressed in the plot. These MSE values are explicitly reported in Table 3 and Appendix E.1. We appreciate the reviewer’s careful observation and have corrected the figure to reflect the full MSE range. **We have also reviewed and refined all other figures to ensure consistent and accurate presentation**.
>
> Importantly, this result underscores a central contribution of our work. As shown in Fig. 2 in Section 4.1 and Fig. 6 in Appendix E.2, the Lotka and Glycolytic systems exhibit nonlinear oscillatory dynamics and are particularly challenging under partial observability. As described in Section 4.2, our Latent MoS model is uniquely designed to capture such structure by learning a mixture of symmetry-preserving latent dynamics, enabling both flexible interpolation and inductive bias. While conventional models (e.g., latent ODE) learn smooth latent flows without exploiting geometric invariances, our approach successfully learns structured latent trajectories that generalize well.

---

> > ### Comment · Reviewer_KeLt · 2025-08-04
> >
> > Thank you for the additional experimental results and clarifications. I will keep my positive rating.

---

> > > ### Author Response · Authors · 2025-08-04
> > > **Follow-Up and Appreciation for Reviewer Feedback**
> > >
> > > **Thank you very much for your encouraging response and for keeping your positive rating**. We truly appreciate your thoughtful suggestions, which helped us improve the clarity of our work. We added sensitivity results showing robust expert selection and clarified the interpolation/extrapolation setup in line with prior work.
> > >
> > > We hope these additions clarify the full scope of the problem and reinforce the state-of-the-art position of our model in addressing it, and further strengthen your confidence in its robustness and contributions.

---

### Official Review · Reviewer_pEQx · 2025-07-02

**Clarity:** 2
**Significance:** 3
**Originality:** 3
**Rating:** 5
**Confidence:** 3

**Summary:**

This paper introduces Latent Mixture of Symmetries (MoS), a continuous-time dynamics model that keeps physical equivariance by treating each latent time-step as a learnable Lie-group action and then blending several such actions with a Mixture-of-Experts gate. The theoretical analysis gives a simple commutation criterion that guarantees any selected action remains equivariant with respect to the known symmetry in the observation space. Experiments on both synthetic ODEs and real-world datasets show improved accuracy and qualitatively symmetry-consistent latents compared with baselines.

**Questions:**

1. It would be helpful to state explicitly which symmetry (or combination) exists in each experimental dataset and provide quantitative evidence that MoS captures it (the PCA in Fig. 2 is a good start, but more discussion would be beneficial).
2. Why is Corollary 1 limited to planar rotation? Did any experiment involve true 3D rotational symmetry, and if so how was it handled?
3. Could the authors clarify how MoS might extend beyond rotation/translation/scaling?
4. How sensitive is performance to the size of the symmetry candidate library? Will more irrelevant experts hinder the performance?

**Ethical Concerns:**

["NO or VERY MINOR ethics concerns only"]

**Final Justification:**

The authors addressed most of my concerns, so I increased my score accordingly.

**Limitations:**

Although the authors list some limitations, the paper would benefit from a failure-case analysis, e.g., examples where the gating network consistently picks the wrong expert and how that might be resolved.

**Quality:**

3

**Strengths And Weaknesses:**

Strength
1. Using MoS to learn a data-driven mixture of symmetry experts is a neat way to model symmetry in dynamical systems with unknown and/or shifting symmetries.
2. The commutation-based design yields interpretable latent operators while still providing provable equivariance guarantees, which helps both sample efficiency and interpretability.

Weakness
1. The paper does not compare with symmetry-discovery baselines that explicitly discover symmetries from data.
2. The paper does not compare itself against the symmetry-breaking literature (methods that gradually relax or modulate equivariance). Can one view the proposed method like breaking the symmetry from the affine group? A discussion here would be helpful.

---

> ### Author Rebuttal · Authors · 2025-07-30
>
> **Q1. Symmetry-discovery baseline**. The reason we didn't include this baseline is that this type of method has strong requirements for data quality, but we mainly test realistic sample-inefficient cases. For example, GAN and SINDy in [Yang et al., 2024] both require many samples. Naturally, these baselines perform worse than other ODE-structured models like Latent ODE. As the reviewer asked, we conducted the experiments using the official code from [Yang et al., 2024] and tested two recent symmetry-discovery methods: EquivSINDy-r (SINDy-based) and EquivGP-r (Genetic Programming-based). They all require LaLiGAN to discover symmetry, which is further used in SINDy or GP as regularization. We report interpolation errors under a 30\% data drop rate. For the Spiral, Glycolytic, and Lotka systems, the MSEs ($\times 10^{-2}$) are as follows: EquivSINDy-r: 2.12, 2.74, 2.42; EquivGP-r: 2.13, 4.26, 1.61, all of which are around 4× to 10× larger than the results achieved by our Latent MoS. Similar trends hold across other datasets.  These findings highlight that existing symmetry-discovery methods based on GAN priors and symbolic regressors (SINDy/GP) are highly sensitive to data quality and quantity. In contrast, our method remains robust under sparse and limited trajectories.
>
>
> **Q2. Symmetry-breaking literature**. Our work identifies symmetry breaking in two orthogonal axes: ** a mixture of groups and a short- or long-time dependence** (see Section 2.1). This breaking changes the restrictive assumption of a fixed affine group. The mixture of groups appears in literature [Kim et al., 2023], and the locally adaptable symmetry breaking is discussed in [Wang et al., 2022; Wang et al., 2023; Ouderaa et al., 2022]. They are independently discussed, but our method addresses both in a unified framework.
>
> To model symmetry breaking, past methods (1) utilize learnable or non-stationary weights [Wang et al., 2022], (2) add soft regularization  [Kim et al., 2023; Finzi et al., 2021], or (3) construct a symmetry-breaking set as auxiliary inputs [Xie \& Smidt, 2024]. Our model adopts method (1) without the need to construct additional terms or sets.
>
> **Q3. Identified symmetries**.
>
> Our paper states that in Spiral, Glycolytic, and Lotka-Volterra systems, the dominant symmetry is rotation × scaling. It is worth noting that the Glycolytic and Lotka-Volterra systems introduce nonlinear distortions due to their inherent dynamics (see Equations (18) and (19) in the Appendix). Fig. 2 shows that latent MoS recovers this symmetry, but latent ODE can't. For other realistic systems, they don't have ground-truth symmetries.
>
> We also have quantitative results. In the Glycolytic system, for example, even when all symmetry experts are included, the model achieves a low MSE of 0.0033. Importantly, the gating weights are highly concentrated: the dominant expert, i.e., a second-order transformation (rotation × scaling), receives a weight of 0.97, while the next highest (scaling × rotation) receives 0.025, and the remaining experts collectively contribute less than 0.005. This concentration indicates that the model identifies the correct symmetry.
>
> We also conducted an additional experiment. We set the latent dimension m=2 and fixed both the encoder and decoder to identity mappings. This avoids the encoder and decoder jointly transforming the latent dynamics into an equivalent—but potentially different from the true governing equation.
>
> Then, for the spiral dataset, Latent MoS recovers a scaling factor of 0.98 and a rotation matrix R = [0.968, -0.248; 0.248, 0.968], which closely matches the true dynamics used in data generation. The average error compared to the ground truth symmetry is less than 0.05, and the gating weight for this expert (rotation × scaling) is 1. This experiment demonstrates that Latent MoS can faithfully recover interpretable symmetry structures (or their equivalent forms) directly from data.
>
> **Q4. Corollary 1 issue**. The restriction to planar rotation arises because general rotations are non-commutative. E.g., a die rotates along the x-axis and then z-axis, or along the z-axis and then x-axis, will cause different results. In contrast, planar rotations (fixed rotation axis) are commutative. The result is rigorously proved in Appendix A.3, meaning that the state should rotate along one axis at least locally to preserve rotation symmetry. If a rotation continuously changes the axis, symmetry is completely removed.
>
> However, our model is not restricted to 2D but focuses on dynamics in higher dimensions along an axis at a period. As shown in Corollary 1, this is achieved by parameterized planar rotation $R_{\theta}$ (define the rotation axis) and a projection matrix $\hat{P}$ (lift the dimension). For example, we include a 10D power event dataset (results in Table 1) where Latent MoS outperforms all baselines and angle rotations are almost synchronized due to physical constraints.
>
> **Q5. Symmetry extension**. While the current implementation focuses on rotation, translation, and scaling, which already form a rich and compositional symmetry basis, we outline multiple ways in which MoS can be extended:
>
> **1. Higher-order compositions of existing symmetries**. Our model already supports second-order multiplicative compositions in Corollary 3 (e.g., scaling × rotation). This formulation naturally extends to higher-order group actions by multiplying additional Lie group matrices, which captures very complicated interactions among different symmetries.
>
> **2. Generalized symmetry via nonlinear encoder–decoder structure**. As shown in Equation (2) and (3), MoS leverages a decomposed symmetry formulation. By the universal approximation theorem (e.g., [38] in the reference), this setup can represent any continuous nonlinear group action. This means MoS can approximate nonlinear symmetry transformations.
>
> **3. Integration of new symmetry types**. In our paper, we have shown a standard procedure to integrate other symmetry transformations into our MoS.
>
> **(1) Parameterize new group actions** (e.g., discrete symmetries, time-warping, reflection, permutation-equivariant transformations) using constrained neural networks or algebraic constructions.
>
> **(2) Add these as new experts in the MoS module**.  Since MoS is structured as a Mixture-of-Experts, each expert can encode a different symmetry family and is trained jointly through the gating mechanism.
>
> This modularity makes the MoS architecture flexible and extensible.
>
> **Q6. Sensitivity analysis**. We performed sensitivity analysis with respect to noise levels, latent dimensions, and the number of open gates. However, we believe it's good to do an analysis with respect to the size of the candidate library. Using the Glycolytic system with a 90\% drop rate as an example, we vary the number of symmetry experts from 1 to 9 and ensure that the correct symmetry (rotation × scaling) is always included. This helps to evaluate the impact of irrelevant experts. The resulting MSEs ($\times 10^{-2}$) were: 0.19, 0.20, 0.20, 0.25, 0.27, 0.28, 0.31, 0.33, and 0.33, respectively. These results show that as more irrelevant experts are added, performance degrades slightly but remains stable. Even with all 9 experts, the MSE is only 0.0033, illustrated in Q3. This demonstrates that although more irrelevant experts may introduce minor degradation, the model is not highly sensitive to the library size due to the gating mechanism.
>
> **Q7. Limitation**. Our ablation study in Section 4.4 indeed reveals how expert selection influences performance. Specifically, including unnecessary experts (e.g., those not aligned with the system’s underlying symmetries) can lead to a little performance degradation, while removing essential ones significantly harms results. The former observation is illustrated using examples in answers to Q3 and Q6.
>
> For the latter observation, for example, in the Glycolytic system, if we remove the rotation expert, the resulting MSE is 0.014, compared to 0.0033 with all experts included (Section 4.4). The gating weights shift toward translation × scaling (0.82) and scaling (0.15), with the remainder distributed across other experts (0.03). This demonstrates a clear failure case: the model attempts to approximate rotational dynamics using incorrect symmetry compositions, but performance suffers. Thus, including the correct symmetry group is critical.
>
> To address such issues, our ablation study motivates that one can perform a validation-based procedure to progressively remove unnecessary experts and compare the resulting MSE. Importantly, the gating mechanism remains valuable in this process: it helps identify the dominant experts within a candidate subset, effectively narrowing down the search space. This makes the validation process more efficient and scalable. Overall, this approach provides a practical and systematic method for expert selection, particularly when symmetry priors are uncertain or only partially known.
>
> Looking forward, this validation-based insight opens the door to automated expert selection. One promising direction is to leverage Neural Architecture Search techniques, such as RL [zoph et al. 2017] or differentiable methods [zhang et al. 2022]. NAS can search over expert combinations and minimize validation MSE. Alternatively, foundation models and LLMs [tsai et al. 2023] can be used to suggest likely symmetry groups based on contextual or domain-specific descriptions of the system. We also envision incorporating Bayesian optimization or meta-learning strategies [shaw et al. 2019] to efficiently guide gating under limited data. While we leave full automation to future work, our findings suggest that expert selection can be both principled and learnable.

---

> > ### Comment · Reviewer_pEQx · 2025-08-04
> >
> > Thank you for the rebuttal! Most of my concerns are resolved and I have increased my scores.

---

> > > ### Author Response · Authors · 2025-08-04
> > > **Thank You for Your Feedback and Score Increase**
> > >
> > > **Thank you very much for your thoughtful engagement and for increasing your scores**, and we truly appreciate it. We're glad the additional experiments and clarifications helped address your concerns.
> > >
> > > Your suggestions guided us to improve the paper's clarity and depth, particularly around symmetry-discovery baselines, symmetry-breaking literature, and sensitivity analysis. We believe these improvements not only strengthen the presentation but also reinforce the broader contribution of our model to learning symmetry-aware dynamics.

---

### Official Review · Reviewer_C3FL · 2025-07-03

**Clarity:** 2
**Significance:** 3
**Originality:** 3
**Rating:** 5
**Confidence:** 2

**Summary:**

The authors propose the Latent Mixture of Symmetries (Latent MoS), a model for learning dynamics by leveraging symmetry as an inductive bias. Unlike traditional approaches, which either impose a priori available symmetries or discover global ones, Latent MoS learns a mixture of local latent symmetries through Lie group transformations. As a result, it preserves local equivariance and allows for time-dependent adaptations. To enable long-term equivariance, the authors propose hierarchically stacking MoS modules. The authors provide a theoretical justification for the model's ability to preserve equivariance. Empirical experiments in physical systems demonstrate improved performance over baselines in terms of interpolation, extrapolation, and sample efficiency.

**Questions:**

- Can you provide qualitative evidence to support that latent symmetries are interpretable?

**Ethical Concerns:**

["NO or VERY MINOR ethics concerns only"]

**Final Justification:**

I have decided to increase my score from borderline accept to accept based on the detailed response of the authors.

**Limitations:**

Yes.

**Paper Formatting Concerns:**

.

**Quality:**

3

**Strengths And Weaknesses:**

Strengths:
- Quality:
  - The authors propose a mathematically rigorous model with formal guarantees of equivariance preservation.
  - Experimental results, with thoroughly studied ablations, on a diverse set of physical systems demonstrate strong performance in low-data settings.
- Clarity:
  - The flow from single-group equivariance to mixtures and temporal hierarchy clarifies the fundamental ideas.
  - In general, the paper is well-written.
- Significance:
  - The MoS formulation provides structured latent dynamics in various settings, including model-based RL and scientific computing.
- Originality:
  - Time-varying mixture of experts for modeling latent symmetry transformations seems like a novel idea.

Weaknesses:
- Quality:
  - The ablation study does not address imperfect gating.
- Clarity:
  - Derivations in section 3, especially 3.2, may benefit from a more intuitive explanation to non-expert readers.
- Significance:
  - Applicability to closed-loop control, planning, RL, etc, has not been empirically tested.
- Originality:
  - The novelty mainly lies in the combination of well-known components.

---

> ### Author Rebuttal · Authors · 2025-07-27
>
> **Q1. Imperfect gating**. Our ablation study in Section 4.4 indeed reveals how expert selection influences performance. Specifically, including unnecessary symmetry experts can lead to a little performance degradation, while removing essential ones significantly harms results.
>
> However, we emphasize that our MoS gating mechanism can effectively select the true experts. For example, in Section 4.4 and for the Glycolytic system, when we include all symmetry experts, the model still achieves a low MSE of 0.0033. Importantly, the learned gating weights are highly concentrated: the dominant expert, a second-order transformation (rotation × scaling), receives a gating weight of 0.97, while other gating weights are negligible (e.g., 0.025 for scaling × rotation and <0.005 for the rest). This indicates that the gating mechanism correctly identifies and prioritizes the relevant symmetry flow despite the presence of extraneous experts.
>
> Moreover, we observe that removing unnecessary experts can further improve performance. When the translational expert is removed (which is unnecessary for Glycolytic dynamics), the MSE improves to 0.001 and the gating weight for rotation × scaling increases to 0.99. This demonstrates that while the gating works well, a simplified expert set can enhance efficiency and robustness. To formalize this process, our ablation study suggests that one can perform a validation-based procedure to progressively remove unnecessary experts and compare the resulting MSE. Importantly, the gating mechanism remains valuable in this process: it helps identify the dominant expert(s) within a candidate subset, effectively narrowing down the search space. This makes the validation process more efficient and scalable. Overall, this approach provides a practical and systematic method for expert selection, particularly when symmetry priors are uncertain or only partially known.
>
> Looking forward, this validation-based insight opens the door to automated expert selection. One promising direction is to leverage Neural Architecture Search techniques, such as RL [zoph et al. 2017] or differentiable methods [zhang et al. 2022]. NAS can search over expert combinations and minimize validation MSE. Alternatively, foundation models and LLMs [tsai et al. 2023] can be used to suggest likely symmetry groups based on contextual or domain-specific descriptions of the system. We also envision incorporating Bayesian optimization or meta-learning strategies [shaw et al. 2019] to efficiently guide gating under limited data. While we leave full automation to future work, our findings suggest that expert selection can be both principled and learnable.
>
> **Q2. Readability**. The old version aims to give rigorous definitions in m-dimensional latent space, which makes it hard for readers without Lie group knowledge. We are happy to improve the readability by
>
> **(1) Architectural overview.** We will add a dedicated paragraph that walks the reader through how the mixture-of-symmetries (MoS) module is deployed.
>
> **(2) Expert interpretation and parameterization.** We will explain each expert with concrete examples (e.g., 3D rotation, translation, etc.). For each, we will describe the Lie-based parameterization and provide references.
>
> **(3) Visual clarification in Fig. 1.** We will revise Figure 1 to include 3D visualizations of symmetry groups. This will allow readers to immediately grasp the role of each expert.
>
> **Q3. Control tests**. As the topic is dynamic learning, we focused on learning tasks. We are glad to provide a control test from our draft of latent MoS for power system frequency control. The task is to stabilize the generator's frequency. Past methods [Cui et al. 2023] assume a second-order ODE model, i.e., swing equations, to model frequency dynamics. However, real-world dynamics are higher-order. Hence, we utilize latent MoS to learn it. In general, we compare model-free proximal policy optimization (PPO), model-based policy optimization (MBPO) with a Gaussian process (MBPO-GP), MBPO with Latent MoS (MBPO-MoS), and MBPO with the swing equation (MBPO-SE). We track the frequency deviation over time. At time $t = 0.5$, our MBPO-MoS reaches $-0.287$, compared to $-0.975$ for PPO, $0.100$ for MBPO-GP, and $-0.499$ for the MBPO-SE. By the final time point $t = 1.5$, our MBPO-MoS stabilizes near zero with a deviation $-0.024$, the lowest among all methods. In contrast, PPO remains highly unstable ($-0.928$), MBPO-GP yields $0.100$, and  MBPO-SE yields $-0.115$. The general cost (integral of frequency deviations) in the control process is $0.081$ (MBPO-MoS), $0.163$ (MBPO-SE), $0.257$ (MBPO-GP), and $0.51$ (PPO). Our learned dynamics in Latent MoS can much better predict the future frequency, largely improving control performance.
>
> **Q4. Novelty**. The paper is not a simple combination of existing methods, as explained below.
>
> **1. Symmetry-preserving experts**:
> Traditional MoE employs generic neural networks as experts, which can't fit the goal of symmetry preservation. We provide formal proofs showing how each expert guarantees equivariance with specifically parameterized experts. This Lie expert has never been discussed before.
>
> **2. Mixture of latent symmetry flows**:
> Prior methods often assume a single symmetry group. In contrast, Latent MoS introduces a mixture model over multiple symmetry groups, significantly expanding the expressiveness of latent dynamics and enabling the model to adapt to complex symmetries.
>
> **3. Multi-level MoS architecture**:
> Our design supports stacked MoS layers across different temporal resolutions, allowing the model to extract both short- and long-term symmetry patterns. This stands in contrast to prior work, which typically assumes fixed or temporally uniform symmetries, and makes our model more suitable for complex and evolving environments.
>
> **4. Connections to safety guarantees**:
> Because each symmetry expert corresponds to a structured Lie group matrix, the latent dynamics are linearly interpretable. This leads to deep connections with control theory and safe reinforcement learning. For instance, in safe RL, forward-invariant safety sets [Emam, et al. 2022] link symmetry and equivariance to safety, which can be efficiently estimated using our latent linear symmetry flows. This potentially bypasses expensive formal verification tools like SMT solvers [Chang et al. 2019]. We are actively pursuing this as part of ongoing research in safe RL.
>
> **5. Significant empirical advances**:
> Across multiple benchmarks, Latent MoS outperforms strong baselines including ODE-based models (Latent ODE, NCDE), sequence models (RNN, ODE-RNN), and transformer-based architectures (Informer, Autoformer). These results validate that symmetry-aware modeling leads to substantial improvements.
>
> **Q5. Interpretability**.
>
> While interpretability is not the primary motivation of our model, it emerges naturally as a byproduct of the linear, Lie group-structured latent dynamics. To support this, we provide both qualitative and quantitative evidence:
>
>
> **1. Qualitative evidence**. As shown in Fig. 2, we visualize the latent trajectories in a 2D setting for three different systems, each dominated by rotation and scaling symmetries (the left column of Fig. 2). These geometric structures are clearly expressed in the latent space learned by Latent MoS. Specifically, our latent trajectories (the right column of Fig. 2) exhibit smooth rotational and scaling motions that align with the known system dynamics. In contrast, the trajectories learned by Latent ODE (the middle column of Fig. 2) lack such structure and are difficult to interpret, highlighting the advantage of our symmetry-preserving formulation.
>
> It is worth noting that the Glycolytic and Lotka-Volterra systems introduce nonlinear distortions due to their inherent dynamics (see Equations (18) and (19) in the Appendix). Despite this, Latent MoS effectively captures the linear symmetry structure in the latent space, while leaving the nonlinearity to the encoder and decoder networks. This result is consistent with the theoretical decomposition described in Section 2.2.
>
>
> **2. Quantitative evidence**. We also provide quantitative support that Latent MoS learns interpretable and symmetry-consistent dynamics in the latent space. In the Glycolytic system, for example, even when all symmetry experts are included, the model achieves a low MSE of 0.0033. Importantly, the gating weights are highly concentrated: the dominant expert, i.e., a second-order transformation (rotation × scaling), receives a weight of 0.97, while the next highest (scaling × rotation) receives 0.025, and the remaining experts collectively contribute less than 0.005. This concentration indicates that the model identifies and relies on the correct symmetry transformation.
>
> To further support this, we conducted an additional experiment motivated by the PCA visualization in Fig. 2, which shows that a 2D latent space suffices to capture the dynamics of the tested systems. We set the latent dimension $m=2$ and fixed both the encoder and decoder to identity mappings. This avoids the encoder and decoder jointly transforming the latent dynamics into an equivalent—but potentially different from the true governing equation.
>
> Then, for the spiral dataset, Latent MoS recovers a scaling factor of 0.98 and a rotation matrix R = [0.968, -0.248; 0.248, 0.968], which closely matches the true dynamics used in data generation. The average error compared to the ground truth symmetry is less than 0.05, and the gating weight for this expert (rotation × scaling) is 1. This experiment demonstrates that Latent MoS can faithfully recover interpretable symmetry structures (or their equivalent forms) directly from data.

---

> > ### Comment · Reviewer_C3FL · 2025-08-09
> >
> > Thank you for addressing my questions and concerns. I have decided to increase my score.

---

### Author Response · Authors · 2025-08-04
**Follow-Up on Rebuttal Responses: Feedback Welcome**

Dear Reviewer,

As the discussion deadline (08/06) approaches, I wanted to kindly follow up to see if our responses addressed your concerns. We’ve made every effort to clarify the technical details, highlight the innovations of our model, and add additional literature and tests. If you have any additional thoughts or feedback, we’d be glad to continue the discussion before the deadline.

Thank you again for your time and consideration.

---

> ### Author Response · Authors · 2025-08-06
>
> Dear Reviewers:
>
> Thank you all for your time and thoughtful feedback. We sincerely appreciate that Reviewers pEQx and KeLt have already responded and **maintained or increased their positive ratings**. Your comments and suggestions have helped us improve the paper.
>
> For the remaining reviewers, as the NeurIPS policy now extends the discussion deadline to Aug 8 (AoE), we kindly wanted to ask whether our rebuttal addressed your concerns. We would be grateful for any additional feedback or clarification you might have. In particular, for Reviewer UuCF, we really appreciate your questions about **time-variant and mixture of group symmetries, which directly echo our core contributions**. These design choices are intentional and crucial for capturing complex real-world dynamics.
>
> Thank you again for your efforts and engagement throughout the review process. We truly value your time and input.

---

> > ### Author Response · Authors · 2025-08-07
> >
> > Dear Reviewers C3FL and UuCF:
> >
> > As the discussion deadline approaches (Aug 8, 11:59pm AoE), I wanted to gently follow up on our previous message. We understand this is a busy time, but we would be very grateful if you could briefly let us know whether our rebuttal and responses have addressed your concerns. Your feedback is extremely valuable. Many thanks again for your time and contributions to this review process.

---

### Note · Authors · 2025-08-12

We sincerely thank all reviewers for their time, effort, and constructive feedback throughout this process. We are grateful that three reviewers have increased their scores after our discussions, and one has maintained a positive rating, reflecting their recognition of the model’s contributions. We deeply appreciate the reviewers’ engagement, as this outcome provides strong evidence of consensus on the significance and quality of our work!

---

### Decision · Program_Chairs · 2025-09-17

**Decision:**

Accept (poster)

**Comment:**

This paper introduces Latent Mixture of Symmetries (MoS), a continuous-time dynamics model that preserves physical equivariance by treating each latent time step as a learnable Lie group action and blending multiple such actions through a Mixture-of-Experts gating mechanism. The strengths of the paper lie in its significance, clarity, and originality, although the evaluations could be further strengthened. The authors have adequately addressed most of the reviewers’ comments, and the work makes a valuable contribution to the field.